# Optical imaging of the intrinsic adsorption kinetics in single zeolite nanoparticles

Xuannuo Yi[1], Haoran Han[1], Aosheng Chang[1], Ziyuan Liu[2], Qingxue Hui[3], Chongqin Zhu [2], Zhaoqiang Zhang [3], Shasha Liu [1,4] ✉ & Wei Wang [1] ✉

The confinement effect is increasingly recognized as a critical factor influencing guest–framework interactions in molecular sieves, yet its impact on adsorption kinetics remains largely unexplored. Conventional ensemble measurements on milligram-scale particle assemblies yield apparent adsorption kinetics that conflate dynamic molecular interactions with macroscopic mass transport. Here, we present an optical imaging approach that quantitatively monitors interaction-dominated adsorption by reducing the sample size to the single-nanoparticle level (sub-picogram scale). The results enable the determination of intrinsic rate constants and activation energy barriers for elementary adsorption and desorption steps. A confinement-induced reversal of adsorption kinetics, relative to proton affinities, is observed among homologous light olefins on the same ZSM-5 nanoparticle. This finding reveals that confinement—rather than interaction strength—primarily governs adsorption kinetics at the single-nanoparticle level and provides a general platform for probing and rationally designing molecular sieves for diverse applications.

Solid porous materials, such as zeolites and molecular sieves, have been widely used in industrial applications since the 1950s, particularly in gas separation and heterogeneous catalysis, due to their well-defined as well as tunable pore structures[1–3]. Beyond simple consideration of size and shape selectivity, the confinement effect of nanopores—where molecules are trapped in a constrained environment—has been increasingly recognized as an essential factor influencing their performance, including reaction pathways and adsorption selectivity[4–6]. To achieve a fundamental understanding of the confinement effect in solid porous materials and to further tailor their behavior in real applications, it is indispensable to characterize molecule-framework interactions experimentally.

Adsorption serves as a key process for studying the confinement effect in molecular sieves, as it represents the first step and a fundamental interaction between confined molecules and the framework[7–9]. Spectroscopic techniques, such as vibrational and absorption spectroscopy, have been widely used to probe the electronic and structural states of target molecules before and after adsorption on nanoporous materials[10–13]. Increased molecular orbital energies compared to their bulk counterparts are observed, providing experimental evidence that supports theoretical predictions, as a result of the restricted mobility and limited structural flexibility of molecules encapsulated in confined nanopores[14]. More recently, state-of-the-art fluorescence microscopy and transmission electron microscopy enabled the direct visualization of single molecules within the framework, offering insights into their spatial distribution and coordination environment at the nanoscale[15–18]. While these results have significantly enhanced our understanding of the confinement effect on adsorption within solid porous materials, they primarily represent static "end-point" characterizations, leaving the influence on the dynamic adsorption kinetics of guest molecules elusive.

[1]State Key Laboratory of Analytical Chemistry for Life Science, Chemistry and Biomedicine Innovation Center (ChemBIC), School of Chemistry and Chemical Engineering, Nanjing University, Nanjing, China. [2]College of Chemistry, Key Laboratory of Theoretical & Computational Photochemistry of Ministry of Education, Beijing Normal University, Beijing, China. [3]State Key Laboratory of Coordination Chemistry, School of Chemistry and Chemical Engineering, Nanjing University, Nanjing, China. [4]Shenzhen Research Institute of Nanjing University, Shenzhen, China. ✉e-mail: ssliu@nju.edu.cn; wei.wang@nju.edu.cn

More broadly, this knowledge gap stems from the inherent complexity of adsorption kinetics within molecular-scale nanoporous materials, where multiscale processes are commonly at play—including macroscopic transport limitations and site-specific adsorption at the nanoscale[5]. This issue, coupled with the conventional dynamic adsorption characterizations typically conducted on bulk-phase samples at the milligram scale, such as adsorption-desorption breakthrough curve analysis, temperature-programmed desorption, and intelligent gravimetric analysis, leads to the intrinsic adsorption kinetics becoming more entangled with macroscale mass transfer in the observed apparent results[19–21]. Fortunately, as suggested by diffusion theory and the zero-length column technique, reducing the sample size offers a promising strategy to minimize mass transfer contributions on experimentally obtained apparent adsorption kinetics[22,23]. The diffusion kinetics of molecules within a single microcrystal have been achieved by Interference microscopy (IFM) and infrared microscopy (IRM), but IFM is limited to observe the nanoporous materials with well-defined morphologies and large crystal sizes (>20 μm), and synthesis of such large crystals is time-consuming and requires complex experimental conditions[22,24,25]. To achieve an accurate characterization and quantitative determination of the intrinsic dynamic adsorption kinetics for studying the confinement effect in molecular sieves, it is therefore highly desirable—yet remains technically challenging and largely unexplored—to further reduce the sample size to a single nanoparticle level (at the picogram level), which represents the fundamental unit of solid porous materials in practical applications.

In the present work, we employed an atmosphere and temperature controllable dark-field microscopy (DFM) to optically image the dynamic adsorption process on single solid porous nanoparticles in situ and in real time, by tracking changes in their scattering intensity as a result of refractive index variations induced by mass density change. A reversible increase and decrease in the scattering intensity of individual Zeolite Socony Mobil-5 (ZSM-5) nanoparticles were clearly observed during the adsorption and desorption of olefins, respectively. Quantitative analysis first confirmed the negligible contribution of mass transfer to the observed adsorption dynamics and subsequently revealed a pseudo-first-order reversible adsorption mechanism. Built on these findings, the intrinsic rate constants for the elementary adsorption and desorption steps were extracted. In addition to providing kinetic insights, the determination of intrinsic rate constants also enabled the calculation of adsorption thermodynamics, with equilibrium constants derived from the ratio of rate constants. In contrast to the adsorption kinetics, the determined equilibrium constants showed good agreement with theoretically calculated protonation energies of the three olefins, validating the reliability of the developed methodology. A reversal trend, compared to their proton affinities, was uncovered in the adsorption rate of light olefins ($C_2H_4$, $C_3H_6$, n-$C_4H_8$) on the very same ZSM-5 nanoparticles. This was further supported by a corresponding reversal in their activation energy barrier heights and a systemical comparison of adsorption kinetics on zeolites with different pore sizes. These findings provide comprehensive insights for understanding and optimizing their performance in catalytic and separation applications, and highlight the necessity of assessing the confinement effect on adsorption kinetics within solid porous materials.

## Results

### Optical imaging of the dynamic adsorption and desorption process of $C_3H_6$ on single ZSM-5 nanoparticles

ZSM-5 was selected as a model solid porous material due to its important role in various industrial catalytic reactions, such as fluidized catalytic cracking and methanol-to-olefins[26]. Among these applications, $C_3H_6$ is one of the most common reactants or products and was thus chosen as the adsorbate in this study[27]. Before the kinetic measurements, the ZSM-5 nanoparticles were comprehensively characterized in terms of their structural features and acidity (Supplementary Fig. 1). Transmission electron microscopy images and X-ray diffraction patterns confirmed that the ZSM-5 nanoparticles exhibited a typical coffin-like morphology with a well-defined MFI framework. The Si/Al ratio was determined to be 52.1 by an inductively coupled plasma optical emission spectrum (Supplementary Table 1). The surface area was determined to be 352 $m^2$/g from the $N_2$ adsorption isotherm using the Brunauer-Emmett-Teller equation. To evaluate the acidity of the used ZSM-5 nanoparticles, ammonia temperature-programmed desorption ($NH_3$-TPD) analysis was performed, revealing two distinct desorption peaks at approximately 199 °C and 433 °C. These results are in good agreement with the literature[28,29], suggesting the structural integrity and good performance of the ZSM-5 nanoparticles.

To optically image the dynamic adsorption and desorption process of $C_3H_6$ molecules on a single ZSM-5 nanoparticle, the DFM was built by integrating a home-designed micro-gas-flow system and a micro-area-heating module into a commercial Ti-2E setup (Supplementary Fig.2), based on our recently developed technique for measuring the $NH_3$-TPD spectra of single zeolite nanoparticles[30]. Briefly, individual ZSM-5 nanoparticles were deposited on a home-fabricated substrate, which serves to both support the ZSM-5 nanoparticles for imaging and regulate their temperature via Joule heating. The substrate was then sealed by a micro-chamber, allowing for precise control of the gas atmosphere around ZSM-5 nanoparticles via a multichannel valve to trigger the adsorption and desorption cycles. Miniaturizing the gas flow and temperature control modules helps to improve their compatibility with the microscope and to stabilize optical imaging performance against thermal and atmospheric fluctuations. During the in-and-out movement of $C_3H_6$ molecules from single ZSM-5 nanoparticles, the adsorption of $C_3H_6$ on a single ZSM-5 nanoparticle was monitored in real time and in situ by the refractive-index sensitive DFM, by taking advantage of the inherent refractive index change of the particle accompanied by the change in the amount of $C_3H_6$ molecules adsorbed on their surfaces. The dynamic process was recorded using a CCD camera. Representative DFM images of ZSM-5 nanoparticles are shown in Fig. 1a and Supplementary Fig. 3, where each bright dot represents an individual ZSM-5 nanoparticle with high contrast to the dark background.

In a typical measurement, the ZSM-5 nanoparticles were preheated at 400 °C for five minutes under a pure $N_2$ flow to remove any potential pre-adsorbed species and to activate their surface for the following adsorption. The temperature was then lowered and kept at 105 °C throughout the measurement. At a certain moment ($t = 0$ s), $C_3H_6$ vapor was introduced into the chamber with a gas flow rate of 3 mL/min. This rate was chosen to balance the effects of gas flow on mass transfer, thermal equilibrium, and stability of the imaging system (Supplementary Fig. 4). During the entire experiment, the camera continuously recorded DFM images at a rate of 1 frame per second. A gradual increase in the optical intensity of the ZSM-5 nanoparticles was immediately observed after switching the gas from $N_2$ to $C_3H_6$ vapor (Fig. 1b), indicating the successful uptake of $C_3H_6$ onto the particle, driven by the interaction between the electron-rich π-bonding orbital of $C_3H_6$ molecules and the vacant orbital of the proton in the ZSM-5 framework. Further in-situ spectroscopic measurements confirmed this interaction occurs on Brønsted acid sites (Supplementary Fig. 5). The increased optical intensity was resulted from the increased mass density and corresponding refractive index of the particle during adsorption, thus reflecting the amount of the adsorbed $C_3H_6$ molecules. This observation is consistent with earlier findings, where guest molecule adsorption induces changes in the refractive index of nanoporous materials, resulting in shifts in their optical interference fringes[22,25]. This point was further supported by the results as follows. i) The optical intensity of ZSM-5 nanoparticles gradually decreased to

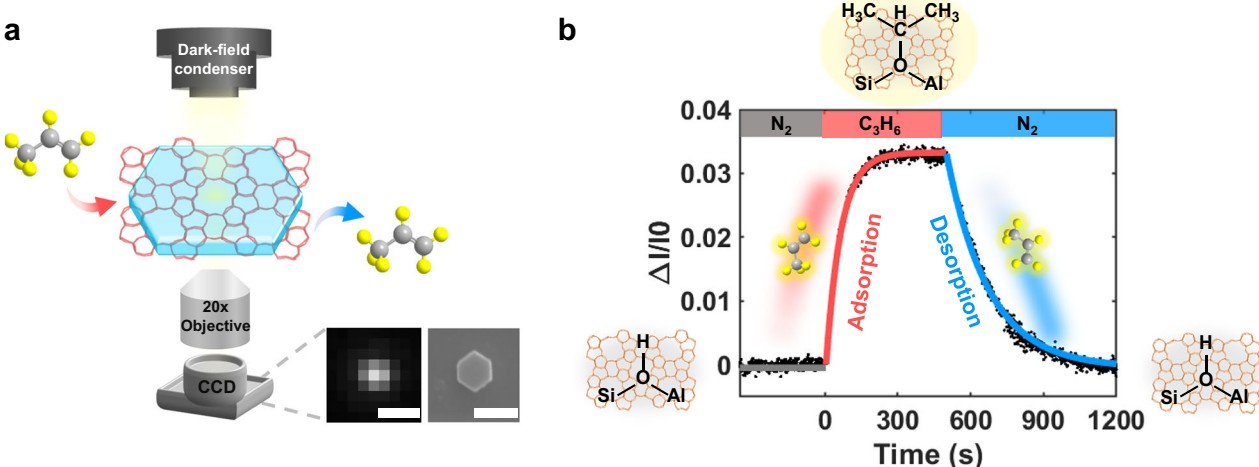

**Fig. 1 | Optical imaging of adsorption and desorption of $C_3H_6$ on single ZSM-5 nanoparticles. a** Schematic diagram of the set-up. The scale bars in both DFM and SEM images are 1 μm. Atom colors: carbon (grey), hydrogen (yellow). **b** Optical response of a representative ZSM-5 nanoparticle during an adsorption-desorption cycle.

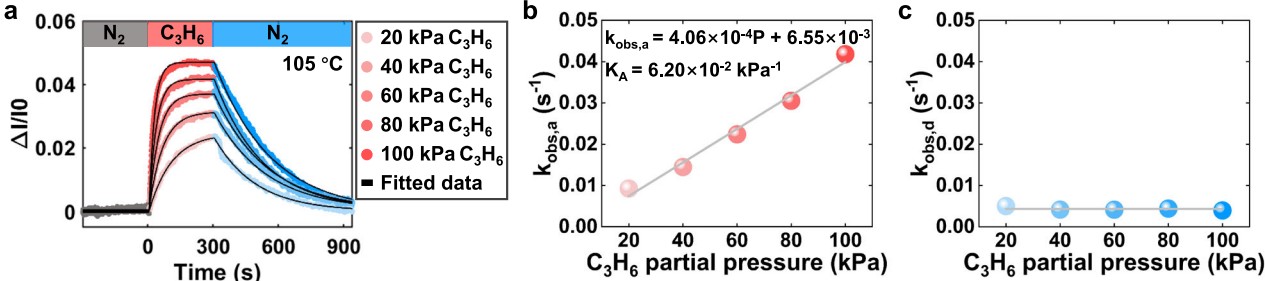

**Fig. 2 | Quantitative assessment of $C_3H_6$ concentration-dependent adsorption/ desorption kinetics. a** The normalized optical intensity curves under different partial pressures of $C_3H_6$. **b** The observed adsorption rate constants linearly increased with the partial pressure of $C_3H_6$. **c** No dependence was investigated between desorption rate constants and the partial pressure of $C_3H_6$.

the original value after switching the gas back to $N_2$. Alternative mechanisms such as thermal decomposition, carbon/pollutant deposition, or surface contamination were excluded, as they would result in irreversible changes. ii) No change in optical intensity was observed in control experiments when acidic $CO_2$ or neutral $O_2$ was used to replace electron-rich $C_3H_6$ (Supplementary Fig. 6), or when non-acidic Si nanoparticles replaced the ZSM-5 nanoparticles (Supplementary Fig. 7). iii) Quantitatively, the observed ~4% change in optical intensity is in excellent agreement with the 1 mmol/g adsorption capacity commonly reported in the literature[31,32]. Note that the change in optical intensity was normalized to the initial intensity of ZSM-5 nanoparticles ($\Delta I/I_0$), enabling a more effective comparison between measurements. Detailed data analysis was provided in the Supplementary Information (Supplementary Fig. 8).

### Quantitative assessment of the intrinsic adsorption and desorption kinetics using the pseudo-first-order reversible chemical adsorption model

To further elucidate the adsorption kinetics at a single nanoparticle level, the adsorption-desorption cycles were performed at varying adsorbate concentrations. As shown in Fig. 2a, the amount of adsorbed $C_3H_6$ molecules on the same individual ZSM-5 particle increases with the partial pressure of $C_3H_6$. Further quantitative analysis suggested that the time course of the adsorbed $C_3H_6$ molecules followed single-exponential growth kinetics across all partial pressures (Fig. 2a and Supplementary Fig. 9). Plotting the growth rate constants ($k_{obs,a}$) as a function of the partial pressure of $C_3H_6$ revealed a linear dependence (Fig. 2b), indicative of a pseudo-first-order adsorption mechanism

herein[33]. By contrast, no adsorbate concentration dependence was observed in the apparent desorption rate constant ($k_{obs,d}$) (Fig. 2c), although it still followed exponential kinetics. This discrepancy was further confirmed by the statistical result of 79 particles (Supplementary Fig. 10), excluding diffusion as a primary contributor to the observed optical response, since the adsorbate concentration should similarly affect both adsorption and desorption under diffusion-limited conditions. We clarify that a continuous flow of $N_2$ was pumped throughout the desorption process to ensure the effective removal of desorbed $C_3H_6$ from the atmosphere surrounding the ZSM-5 particle.

To further validate our hypothesis, we examined the dependence of adsorption rate constants on particle size, as distinct correlations would arise in diffusion-dominated and interaction-controlled kinetics. Taking advantage of the high throughput of the developed technique, the adsorption behaviors of tens of ZSM-5 particles with varying sizes were assessed in a single experiment. In line with the prediction, no significant size dependence was found in either $k_{obs,a}$ or $k_{obs,d}$ (Supplementary Fig. 10). Interestingly, under the very same measurement condition, a negative correlation gradually emerged when the acidity of particles was reduced by increasing the Si/Al ratio in their structure (maintaining the same MFI framework), particularly in non-Al silicalite-1 particles (Supplementary Fig. 11 and Supplementary Fig. 12). These findings suggest that the strong acidity of ZSM-5 nanoparticles leads to $C_3H_6$-acid site interactions that dominate the adsorption kinetics, thereby providing an effective model for studying intrinsic adsorption and desorption kinetics under confinement.

Given the adsorption process is reversible and essentially governed by the competition between elementary adsorption and

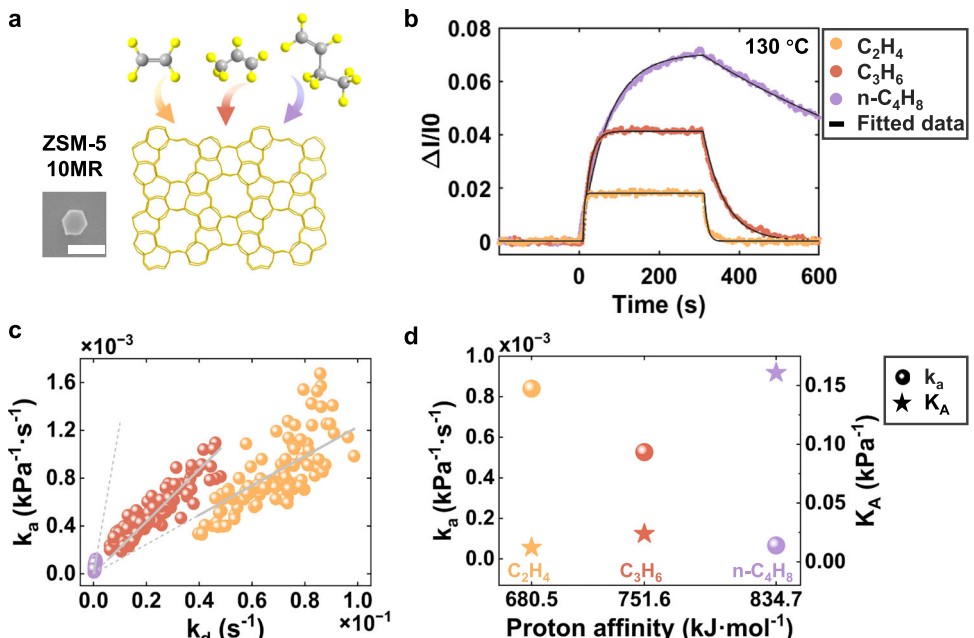

**Fig. 3 | In situ comparison of the adsorption kinetics and thermodynamics of homologous light olefins ($C_2H_4$, $C_3H_6$, n-$C_4H_8$) on the same individual ZSM-5 nanoparticles. a** Schematic diagram of the measurement. The scale bar is 1 μm. Atom colors: carbon (grey), hydrogen (yellow). **b** Optical adsorption-desorption curves of the three olefin molecules measured on a representative ZSM-5 nanoparticle. The black lines represent the fitting results. **c** The statistical linear dependence between the rate constants for the elementary adsorption and desorption steps of each olefin molecule. $R^2_{C2H4} = 0.84$, $R^2_{C3H6} = 0.97$, $R^2_{n-C4H8} = 0.95$. **d** The experimentally determined $k_a$ (left axis) and $K_A$ values (right axis) of three olefins as a function of their theoretically calculated protonation energies.

desorption steps, the obtained $k_{obs,a}$ was accordingly derived as the sum of rate constants for the two elementary steps: $k_{obs,a} = k_a \times P_{C3H6} + k_d$, where $k_a$ ($kPa^{-1} \cdot s^{-1}$), $k_d$ ($s^{-1}$) and $P_{C3H6}$ (kPa) represent the elementary adsorption rate constant, desorption rate constant and partial pressure of $C_3H_6$, respectively. More details about the derivation can be found in the supplementary information. The intrinsic $k_a$ and $k_d$ were quantitatively determined as $4.1 \times 10^{-1}$ $kPa^{-1} \cdot s^{-1}$ and $6.6 \times 10^{-3}$ $s^{-1}$, for the first time, from the transient kinetic measurements of adsorption processes at a single ZSM-5 nanoparticle level. Note also that the $k_d$ value deduced from the y-intercept ($6.6 \times 10^{-3}$ $s^{-1}$, Fig. 2b) closely matches that obtained from direct measurements ($k_{obs,d}$, $4.4 \times 10^{-3}$ $s^{-1}$, Fig. 2c). This consistency suggests the reliability of both the measurement and the analysis.

In addition to providing insights into the adsorption kinetics and mechanisms, the developed methodology also enables the assessment of adsorption thermodynamics. This is because the equilibrium association constant ($K_A$), a key parameter proportional to adsorption affinity, could be calculated as the ratio of the elementary adsorption and desorption rate constants, i.e., $K_A = k_a/k_d$[34,35]. A consistent slope between $k_a$ and $k_d$ was observed across tens of particles exhibiting very different adsorption kinetics, following a Gaussian distribution with an average $K_A$ value of $2.38 \times 10^{-2}$ $kPa^{-1}$ and a standard deviation of $5.10 \times 10^{-3}$ $kPa^{-1}$ (Supplementary Fig. 13). It is worth noting that no equilibrium is required to determine adsorption thermodynamics from transient kinetics experiments, which represents a strength of this approach, particularly in cases where equilibrium cannot be reached or is difficult to achieve.

### Exploration of the confinement effects on olefin adsorption kinetics via single particle intrinsic kinetic analysis

Based on the developed methodology, we further studied the confinement effect on the adsorption kinetics by comparing the adsorption behaviors of three homologous light olefins ($C_2H_4$, $C_3H_6$, n-$C_4H_8$) on the very same single ZSM-5 nanoparticle (Fig. 3a and Fig. 3b). One significant difference is that the change in optical intensity linearly increased with their molecular weights, and the ratio closely mirrored the relative molecular weights of the three molecules, suggesting that all three light olefin molecules can effectively access the active sites on the surface of ZSM-5 nanoparticles. This is sensible as their kinetic diameters are all smaller than the pore size of ZSM-5 nanoparticles[36,37]. Secondly, the experimentally determined $K_A$ values of the three adsorbates (Fig. 3c)—obtained from the statistical linear dependence between their elementary adsorption and desorption rate constants across 103 ZSM-5 particles—exhibited a positive correlation with the theoretically calculated gas-phase protonation energies (Fig. 3d)[38–41], further validating the reliability of the developed technique. It is noteworthy that no size dependence was observed in the adsorption rate constants of any of the three olefins, suggesting that the comparison herein is effectively based on intrinsic adsorption and desorption kinetics (Supplementary Fig. 14). Thirdly, a trend was observed when comparing the intrinsic adsorption kinetics of the three olefins with their equilibrium association constants: both the adsorption and desorption rate constants decrease as proton affinity increases (Fig. 3d and Supplementary Fig. 15). This behavior represented a reversal of the trend in the gas phase, where molecules with higher proton affinities tend to bind more rapidly to the same acid sites[42]. These findings highlighted the confinement effect, rather than interaction strength, as the primary factor governing the adsorption kinetics of olefins in single zeolite nanoparticles.

The observation of confinement-induced reversal of the adsorption kinetics (CIRAK) was further validated from two complementary perspectives. First, the activation energy barriers for the elementary adsorption and desorption steps of these three olefins on ZSM-5 nanoparticles were estimated and compared, since these barriers serve as another key characteristic for understanding the adsorption kinetics under confinement. Figure 4a demonstrates that both the adsorption and desorption rates of $C_3H_6$ increase with temperature, and a linear temperature dependence was further observed. The activation energy barriers for adsorption (Fig. 4b, $E_a$) and desorption (Fig. 4c, $E_d$) were subsequently determined from their slopes according to the Arrhenius

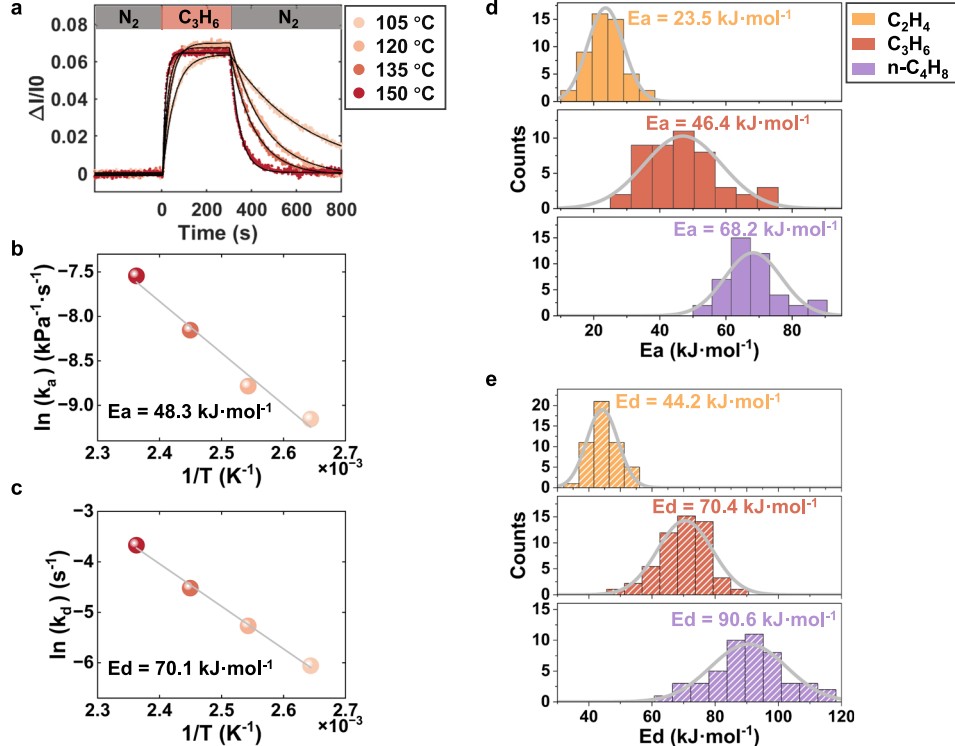

**Fig. 4 | Determination on the activation energies for adsorption and desorption of three olefins. a** The normalized optical intensity curves under different temperatures of $C_3H_6$. **b, c** Determination on the adsorption and desorption activation energies of a single ZSM-5 particle from the linear dependence according to the Arrhenius equation. **d, e** The histograms of the adsorption ($E_a$) and desorption ($E_d$) activation energies of 50 ZSM-5 particles for $C_2H_4$, $C_3H_6$, n-$C_4H_8$, respectively.

equation. The results for $C_2H_4$ and n-$C_4H_8$ are shown in Supplementary Fig. 16 and Supplementary Fig. 17, respectively. Note that the nearly constant adsorption amounts observed at all temperatures are possibly due to the relatively narrow experimental temperature range (Supplementary Fig. 18). The reproducibility of the extracted kinetic parameters was also confirmed by the identical results obtained from repeatable measurements (Supplementary Fig. 19). By counting the $E_a$ and $E_d$ values for tens of particles in the field of view, the statistical results clearly demonstrate that both barrier heights for elementary adsorption (Fig. 4d) and desorption (Fig. 4e) of light olefins increased with the proton affinity of the adsorbates. The activation energy distributions reflects the inter-particle heterogeneity within the same batch of ZSM-5 particles. Specifically, n-$C_4H_8$, which has the highest affinity with acid sites, also has the highest barrier height to overcome for adsorption, thus with the slowest adsorption rates as shown in Fig. 3. Accordingly, we speculate that the observed reversal in adsorption kinetics induced by the confinement effect originates from the spatial constraints within the zeolite framework, which limit the structural flexibility of olefin molecules. As a result, the adsorption process requires passing through a transition state with a higher activation energy compared to that in free space. This effect is more pronounced for larger molecules, leading to the inverse trend in adsorption kinetics relative to their proton affinities.

On the other hand, we systematically investigated the adsorption behaviors of the three olefin molecules on zeolites with a range of pore sizes, including those both smaller and larger than that of ZSM-5 (Supplementary Table 2). The results indicated that the reversal between adsorption kinetics ($k_a$) and thermodynamics ($K_A$) became progressively weaker with increasing pore size, directly evidencing the critical role of pore size in the observed CIRAK. Specifically, identical measurements were performed on SSZ-13—a representative small-pore CHA-type zeolite with a three-dimensional pore system featuring 8-membered-ring (8MR) windows, whose pore size is smaller than that of

10MR windows in ZSM-5 (Supplementary Fig. 20 and Supplementary Table 2). Single nanoparticle kinetic measurements showed that $k_a$ maintained a negative correlation with the proton affinity of the three olefins in SSZ-13 (Fig. 5a). In contrast, CIRAK was absent for HY zeolite, which possesses a 12MR pore system with a larger aperture. In this case, both $k_a$ and $K_a$ exhibited positive correlations with proton affinity (Fig. 5b). Moreover, analogous CIRAK behavior was observed in the adsorption of light alcohols on ZSM-5 (Supplementary Fig. 21). These results not only validate our hypothesis but also highlight the broad applicability of CIRAK in the adsorption of diverse adsorbates on zeolites with distinct framework structures.

## Discussion
In summary, we developed an optical imaging methodology to monitor the intrinsic adsorption process on a single ZSM-5 nanoparticle in a quantitative, high-throughput, and in situ manner. By controlling the experimental conditions, single nanoparticle measurements enable direct observation of interaction-dominated adsorption dynamics and allow for the quantitative determination of intrinsic rate constants and the activation energy barriers for both elementary adsorption and desorption steps. Taking advantage of the highly informative transient kinetic measurements and the broad applicability of the refractive index-sensitive DFM, the adsorption kinetics and thermodynamics of various olefins were studied and compared on the very same individual ZSM-5 nanoparticle, uncovering that the adsorption of light olefins follows pseudo-first-order reversible kinetics, with the adsorption rate constants exhibiting a reversal trend relative to their proton affinities. These observations highlight the importance of studying the confinement effect on adsorption kinetics within solid porous materials.

Optical imaging techniques, represented by surface plasmon resonance, have yielded valuable insights into biomolecular interactions since the 1980s, with the binding kinetics they provide becoming the gold standard in mechanistic study and drug screening in

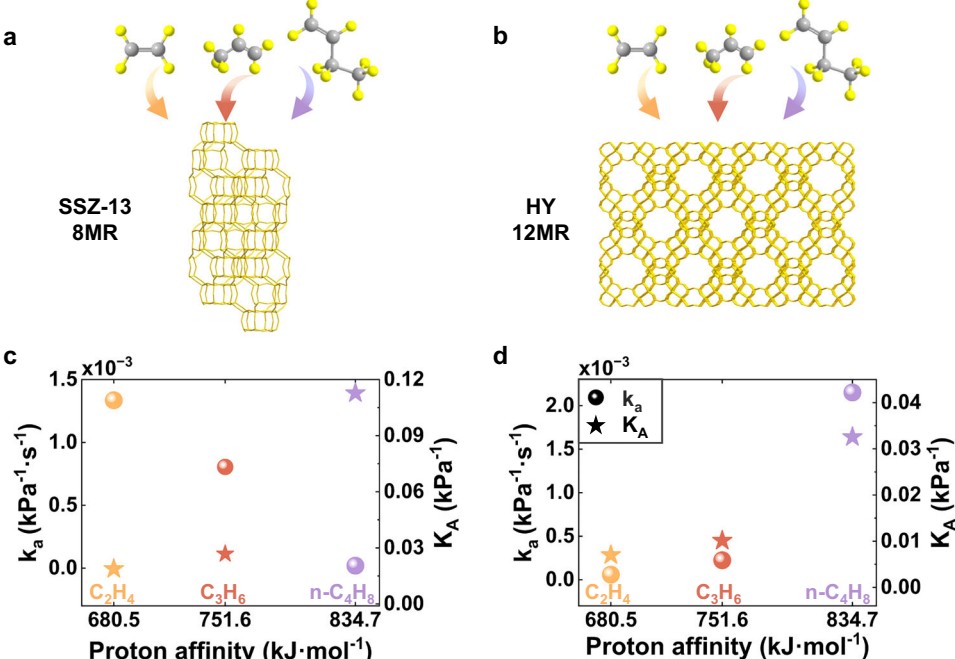

**Fig. 5 | Confinement effect on the adsorption kinetics and thermodynamics of light olefins on zeolites with different pore sizes. a, b** Schematic diagrams of the measurements of SSZ-13 and HY zeolite. Atom colors: carbon (grey), hydrogen (yellow). **c, d** The experimentally determined $k_a$ (left axis) and $K_A$ values (right axis) of three olefins as a function of their theoretically calculated protonation energies.

biopharmaceutical industry[43–46]. The characteristic biomolecular interaction—monolayer self-assembly at the surface—naturally minimizes the influence of mass transfer on the obtained adsorption kinetics. By contrast, this study overcomes mass transfer limitations in zeolites by technically combining optical microscopy and the single nanoparticle measurement strategy, marking the beginning of kinetic studies between molecules and porous materials in chemical industry. We foresee that the developed methodology will advance intrinsic adsorption kinetic measurements, enabling more efficient and targeted applications of solid porous materials in catalysis, separation and other industrial processes.

## Method
### Materials
The ZSM-5 (Si/Al = 50 and 400), silicalite-1, SSZ-13 (Si/Al = 40) and HY (Si/Al = 50) zeolite particles used in this work were purchased from the Catalyst Plant of Nankai University (Tianjin, China). All gases used in this work, including $N_2$, $C_2H_4$, $C_3H_6$ and n-$C_4H_8$, have a purity of above 99.99% and were purchased from Nanjing Tianze Gas Co., Ltd. The Si nanoparticles were purchased from Shanghai Aladdin Biochemical Technology Co., Ltd.

### Materials characterization
The crystallographic structure of the used ZSM-5 powder was characterized by X-ray diffraction (Shimadzu XRD-600) with monochromatic Cu Kα1 radiation. The morphology of the ZSM-5 catalysts was investigated via a field-emission scanning electron microscope (SEM, Phenon) with a secondary electron detector and a transmission electron microscope (TEM, JEOL-F200) with accelerating voltages of 200 kV. The temperature-programmed desorption spectrum was characterized on a multipurpose dynamic adsorption instrument with a thermal conductivity detector (TP-5076, Tianjin Xianquan Industry and Trade Development Co., Ltd), where the sample mass was 50 mg and $NH_3$ was used as the probe molecule. The samples were pretreated at 600 °C in a helium gas for 1 h, then 20 vol% $NH_3$ in helium was introduced at 100 °C for 30 min. After adsorption, the samples were purged with helium to remove the physically adsorbed ammonia. The desorption of $NH_3$ was conducted from 100 °C to 600 °C in the helium flow at a heating rate of 10 °C/min. The Brunauer-Emmett-Teller surface area was measured by $N_2$ adsorption-desorption methods with a Micromeritics gas adsorption analyzer (ASAP 2460). The bulk Si/Al ratios were determined using an inductively coupled plasma-optical emission spectrometer (ICP-OES, Agilent 720ES). The in situ infrared spectrum was recorded by an infrared fourier transform spectroscopy (Bruker, INVENIO S).

### Sample preparation
50 mg ZSM-5 powder was dispersed into 5 mL deionized water (18.2 MΩ·cm, Thermo Fisher) and sonicated for 5 min to prevent nanoparticle aggregation. A 20 µL suspension was then dropped onto a coverslip, where ZSM-5 nanoparticles adhered to the substrate via electrostatic interaction. The solution was removed after the substrate achieved a suitable surface coverage (-100 particles/100 × 100 µm²). The sample was dried in a vacuum oven for 12 hours to remove the water adsorbed on the surface of ZSM-5 particles. Before the optical measurement, the sample was sealed with a coverslip using a 1.5 mm-thick polytetrafluoroethylene spacer to form a microchamber, which was compatible with the working distance of the dark-field condenser. Also, pre-fabricated channels on both sides of the spacer enabled connection to a gas switching system for controlling the microchamber atmosph

### Detailed experimental set-up of optical measurements
The experimental set-up consists of three main components.

Gas switching system: This module was used to control the atmosphere of the micro-chamber surrounding the ZSM-5 nanoparticles. A syringe pump (LongerPump, LSP02-1B) was used to control the gas flow rate, and a six-port valve (Cheminert, C25Z-3186D) was introduced to switch different gases smoothly.

Optical imaging system: This module was used to monitor the change of the optical refractive index (mass) of single ZSM-5 nanoparticles during adsorption and desorption processes. The DFM was

built on an inverted commercial optical microscopy (Ti-2E, Nikon) with a dark-field condenser (N.A. = 0.80–0.95) and a 20X objective (N.A. = 0.45). A white LED was used as the illumination source, and a CCD camera (Pike F-032B, Allied Vi-sion Technologies) was used to record the optical images during the entire process with a frame per second of 1.

Temperature control system: This system was designed to control the temperature of the ZSM-5 nanoparticles. A substrate covered with a 50 nm-thicked gold layer served as a resistor to heat the particles sitting on its surface via Joule heating effect. The pattern of the gold film was designed to limit the effective heating area to $500 \times 500 \ \mu m^2$ to reduce the influence of heating on the stability of the optical imaging system. The voltage applied on the resistor was diiven by a voltage source (CS120, Corrtest).

## Data Availablity
All data supporting the findings of this study are available within the main manuscript or Supplementary Materials. All data are available from the corresponding author upon request. The data that support the findings of this study are provided in the Source data file. Source data are provided with this paper.

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

## Acknowledgements

We acknowledge the financial support from the National Key Research and Development Program (No. 2024FYA1509600, W.W.) and National Natural Science Foundation of China (Nos. 22327803, W.W.; 22304078, S.L.), Natural Science Foundation of Jiangsu Province (BK20230774, S.L.), Jiangsu Funding Program for Excellent Postdoctoral Talent (2024ZB205, S.L.), Guangdong Basic and Applied Basic Research Foundation (2024A1515011331, S.L.) and the Postgraduate Research & Practice Innovation Program of Jiangsu Province for financial support (KYCX25_0175, X.Y.).

## Author contributions

W.W., S.L. and X.Y. designed the research. X.Y. carried out the experiments. X.Y., S.L. and W.W. analyzed the data and wrote the draft. H.H. and Q.H. helped to perform part of the bulk characterizations. A.C., Z.L., C.Z. and Z.Z. participated in discussing the manuscript. W.W. conceived and supervised the research. All authors commented on the manuscript and approved the final version.

## Competing interests

The authors declare no competing interests.
