## [Transparent Peer Review file · Nature Communications]

Optical imaging of the intrinsic adsorption kinetics in single zeolite nanoparticles

Corresponding Author: Professor Wei Wang

Version 0:

Reviewer comments:

Reviewer #1

(Remarks to the Author)

The manuscript reports a label-free study using dark-field microscopy (DFM) single-particle imaging to quantify the intrinsic adsorption and desorption rate constants, as well as activation energies, of light olefins on ZSM-5, a type of molecular sieve. The integration of DFM under controlled atmosphere/temperature with microfluidic delivery enables in situ and real-time monitoring of single-particle adsorption–desorption events. Based on the observed unusual results, the authors introduce the concept of confinement-induced reversal of adsorption kinetics (CIRAK). The study is innovative, the experimental design is carefully thought out, and the data support the conclusion. Overall, the work makes a valuable contribution to the mechanistic understanding of adsorption in zeolites and could be broadly applied to other types of molecular sieve materials. The following specific questions/suggestions may further improve the manuscript's quality.

The authors report that for three different olefins probed on the same particle, both k_a and k_d decrease with increasing molecular size, while the equilibrium constant K_A continues to increase. This is the most interesting result of this study, and the hypothesis that pore confinement dominates the effective rates is reasonable, but without direct experimental data to support it. To prove the hypothesis that the observed reversal in adsorption kinetics is induced by the confinement effect originating from the spatial constraints within the zeolite framework, is it possible to perform the same experiment in a zeolite nanoparticle with a larger pore size? A less reversal effect should be expected in this case.

Some aspects of the Arrhenius analysis are not entirely clear. The authors point out that with increasing affinity, both E_a and E_d increase. Under fixed partial pressure, exothermic adsorption should normally lead to lower equilibrium coverage at higher temperatures, so the plateau signal would be expected to vary with T .

In Figures 4a, S15a, and S16a, the maximum responses (R_{max}) at different temperatures were pretty close. Is this because of the normalization or because the reaction equilibrium is not temperature-dependent (entropy-driven reaction)?

Were the temperature series collected in a single continuous run or in multiple independent experiments? This information is important because, for C_2 olefins, high temperatures could lead to mild polymerization/coking or gradual site deactivation. Such effects may cause slow drifts in k_a and k_d , which would influence both the slopes (E_a , E_d) and intercepts (A) of the Arrhenius plots. A discussion of how such potential artifacts were controlled or excluded would strengthen the conclusions.

Reviewer #2

(Remarks to the Author)

The manuscript entitled “Optical imaging of the intrinsic adsorption kinetics in single zeolite nanoparticles” reports a novel dark-field microscopy–based approach that enables quantitative measurement of intrinsic adsorption kinetics at the level of single ZSM-5 nanoparticles. The authors extracted adsorption/desorption rate constants and activation barriers, and further revealed a confinement-induced reversal of adsorption kinetics. This work presents a highly innovative methodology with important implications for adsorption studies in porous materials. However, the mechanistic interpretation and supporting evidence require significant strengthening. I therefore recommend major revision, with additional validation and a more rigorous discussion, before the manuscript can be considered for publication in Nature Communications.

1. The change in scattering intensity is fully attributed to refractive index variation upon adsorption. Additional validation

(e.g., in situ IR/Raman spectroscopy or mass spectrometry) would help exclude contributions from surface contamination, thermal effects, or other artifacts.

2. The pseudo-first-order reversible model may oversimplify the adsorption process. The possibility of multi-site adsorption, diffusion–reaction coupling, or site heterogeneity should be more thoroughly discussed, along with the limitations of the adopted model.

3. The claim that confinement effects dominate adsorption kinetics is mainly supported by activation energy statistics and qualitative reasoning. The authors should consider providing additional support, such as molecular simulations (MD/DFT) or complementary spectroscopic evidence. Alternative explanations, such as acid site distribution or channel heterogeneity, should also be discussed.

4. About the generality of the conclusions: The study is limited to ZSM-5 and three light olefins. The applicability of the findings to other adsorbates or zeolite frameworks remains unclear. The authors should either provide broader data or explicitly delimit the scope of their conclusions.

Reviewer #3

(Remarks to the Author)

Reviewer #4

(Remarks to the Author)

The confinement effect has been well accepted in zeolite chemistry, as it can significantly influence the adsorption of adsorbates and overall catalytic performance in catalysis. In this work, Yi et al. correlate adsorption kinetics with the confinement effect in zeolites by employing optical imaging techniques to observe single ZSM-5 nanoparticles. By monitoring optical intensity changes, they assess the intrinsic adsorption and desorption kinetics. Overall, this is an interesting study; however, several major concerns should be addressed as follows:

1. There is a critical lack of clarity and completeness regarding the characterization of the zeolite used in this work. The author states that three distinct materials were employed: ZSM-5 (with Si/Al ratios of 50 and 400) and silicalite-1. However, the authors state that only "the ZSM-5 nanoparticles" were characterized (Fig. S1), without specifying which Si/Al ratio this refers to. Besides, the acidity of those three samples should be evaluated, for example, the Brønsted acid site and Lewis acid site. Probably, the solid-state NMR measurements can be conducted to provide more detailed information.

2. The authors state that the NH₃-TPD spectrum shows peaks "at around 200°C and 400°C". However, upon inspection of Fig. S1c, the second desorption peak appears to be located at a significantly different temperature (approximately 430°C). The text should be revised to accurately reflect the data presented in the figure. Besides, how much acid site in this zeolite? Is there any quantitative relationship between the acid site and the adsorbates adsorption?

3. A significant discrepancy exists between Figure 1b and Supplementary Figure S5, which purportedly show the same phenomenon of the adsorption and desorption of C₃H₆ on ZSM-5. The magnitude of the optical intensity change (the y-axis value) is markedly different between the two figures. In Fig. 1b, the signal change appears to be approximately 0.05, while in Fig. S5 (yellow curve), it appears to be around 0.035.

4. A critical clarification is required regarding the fundamental process under investigation. The manuscript consistently describes the interaction of C₃H₆ with Brønsted acid sites as a "reaction" (e.g., in the context of Fig. 1b). However, the experimental data presented (the reversible optical intensity change in Fig. 1b and the subsequent desorption in TPD profiles) unequivocally characterize a physical/chemical adsorption event, not a chemical reaction. Besides, what's the effect of the Lewis acid site and the defect sites (in both the outside and inner surface) effect on the adsorption effect?

5. The data in Supplementary Figure S11d appears to directly contradict the interpretation provided for the ZSM-5 system and instead points towards a diffusion-controlled process for silicalite-1. The authors show a clear negative correlation between the adsorption rate constant of C₃H₆ and the particle size in silicalite-1. This is a classic signature of a diffusion-limited process: in larger particles, molecules have to travel a longer average path to access the internal pore network, leading to a slower observed adsorption rate.

6. L242-245: The claim that the increase in optical intensity with molecular weight suggests "effective access" to active sites is not logically sound. The optical signal is primarily a function of the adsorbed mass. A heavier molecule will naturally produce a larger signal per molecule, regardless of diffusion or accessibility limitations. The authors should either provide additional data to support the accessibility claim (e.g., adsorption capacities) or rephrase the conclusion to state that the data confirms adsorption occurs, with the signal intensity being mass-dependent.

7. Regarding the analysis in Fig. 3c, the manuscript states that the equilibrium constants (K_A) were determined from the "statistical linear dependence" between elementary adsorption and desorption rate constants. However, the actual K_A values and the crucial statistical parameter (e.g., the coefficient of determination, R²) for this linear fit are not provided either in the figure, the caption, or the main text.

8. In this work, the authors claim that it is the confinement effect but the interaction strength works as the main factor to control the adsorption kinetics of adsorbates. Personally, I don't agree this statement, Confinement is not a standalone steric effect but a holistic one that includes how acid strength manifests in tight spaces. In other words, confinement effect is an overall effect, which contains the contribution of acid strength for adsorbates adsorption. Besides, only one type zeolite was adopted in this work, the different topology zeolite with different channel size should be involved for a systemic comparison. Minor error:

1. In Supplementary Figure S4, the unit for milliliter is written as "ml". The correct SI symbol should be "mL".

Version 1:

Reviewer comments:

Reviewer #1

(Remarks to the Author)

The revision has addressed all questions with new data and additional discussions. I think it is ready to be accepted.

Reviewer #2

(Remarks to the Author)

Comment:

I appreciate the authors' thorough and detailed responses to my original comments. The authors have made substantial improvements to the manuscript. The additional experiments, particularly the systematic comparison across zeolites with different pore sizes (SSZ-13, ZSM-5, HY) and phosphotungstic acid, provide compelling evidence and significantly strengthen the manuscript. However, before final acceptance, I would like the authors to address the following remaining concerns:

1. NaOH etching experiment design

The authors performed NaOH etching experiments to investigate the effects of Lewis acid sites and defect sites. However, I noticed that after alkaline treatment and water washing, the samples were directly dried and measured without ion-exchange and calcination steps. This means the measured samples are Na-form ZSM-5 rather than H-form ZSM-5, as the Brønsted acid sites have been occupied by Na⁺ cations. The correct procedure should include: NaOH etching → water washing → NH₄⁺ ion exchange → calcination → H-ZSM-5. The current experimental design confounds the effects of acid site loss with defect introduction, which likely explains the complex and inconclusive results. I suggest the authors provide control experiments with proper ion-exchange and calcination steps.

2. Source of protonation energy data

The authors use "theoretically calculated protonation energies" as a key parameter to correlate adsorption kinetics and thermodynamics (Figure 3d, Figure 5c-d, etc.). However, the source and calculation method of these values are not specified. The authors should note that in confined zeolite environments, actual protonation/adsorption energies may significantly deviate from gas-phase values due to geometric constraints, van der Waals interactions, and local acid site environments. I recommend the authors clearly state the source and calculation method of the protonation energy data. If possible, provide DFT calculations of adsorption energies and transition state barriers within the confined zeolite channels, which would greatly strengthen the mechanistic interpretation.

3. Discussion of crystal diversity and activation energy distribution

The histograms in Figure 4d-e show substantial broadening in the activation energy distributions across 50 ZSM-5 particles (e.g., E_a for C₃H₆ ranges from ~25 to ~70 kJ/mol). This particle-to-particle heterogeneity is consistent with the "crystal diversity" phenomenon reported by Saint Remi et al. (Nat. Mater. 15, 401-406, Ref. 22). I suggest the authors discuss the possible origins of this activation energy broadening and analyze how this inter-particle heterogeneity affects the reliability of the conclusions.

Once the above concerns are adequately addressed, I recommend acceptance of this manuscript for publication in Nature Communications.

Reviewer #3

(Remarks to the Author)

Reviewer #4

(Remarks to the Author)

The authors have addressed my concerns, I am satisfied with the revisions and recommend its publication in Nature Communications.

Version 2:

Reviewer comments:

Reviewer #2

(Remarks to the Author)

The authors have adequately addressed all previous concerns and the revised manuscript. Now I recommend acceptance for publication.

Responses to Reviewers' Comments

Reviewer #1 (Remarks to the Author):

Comments:

The manuscript reports a label-free study using dark-field microscopy (DFM) single-particle imaging to quantify the intrinsic adsorption and desorption rate constants, as well as activation energies, of light olefins on ZSM-5, a type of molecular sieve. The integration of DFM under controlled atmosphere/temperature with microfluidic delivery enables in situ and real-time monitoring of single-particle adsorption–desorption events. Based on the observed unusual results, the authors introduce the concept of confinement-induced reversal of adsorption kinetics (CIRAK). The study is innovative, the experimental design is carefully thought out, and the data support the conclusion. Overall, the work makes a valuable contribution to the mechanistic understanding of adsorption in zeolites and could be broadly applied to other types of molecular sieve materials. The following specific questions/suggestions may further improve the manuscript's quality.

Response: We appreciate this reviewer for acknowledging the novelty of our work and for the valuable comments that helped us to improve the quality of the manuscript. Detailed point-by-point responses are listed as follows.

The authors report that for three different olefins probed on the same particle, both k_a and k_d decrease with increasing molecular size, while the equilibrium constant K_A continues to increase. This is the most interesting result of this study, and the hypothesis that pore confinement dominates the effective rates is reasonable, but without direct experimental data to support it. To prove the hypothesis that the observed reversal in adsorption kinetics is induced by the confinement effect originating from the spatial constraints within the zeolite framework, is it possible to perform the same experiment in a zeolite nanoparticle with a larger pore size? A less reversal effect should be expected in this case.

Response: We appreciate this reviewer for the valuable suggestions. Accordingly, we have conducted new experiments to systematically examine the adsorption behaviors of ethene (C_2H_4), propene (C_3H_6), and n-butene ($n-C_4H_8$) on zeolites with a range of pore sizes, including those both smaller and larger than the zeolite investigated in the original manuscript (Table R1). An overview of the results is presented in Figure R1, which shows good agreement with this reviewer's comments that the reversal between adsorption kinetics (k_a) and thermodynamics (K_A) becomes progressively weaker as the pore size increases. A detailed discussion is provided below.

Table R1. The structural information of the samples used in adsorption-desorption experiments.

Sample	Framework Type	Featured number of rings (MR)	Pore size (nm)
SSZ-13	CHA	8 MR	0.38
ZSM-5	MFI	10 MR	0.51×0.55 0.53×0.56
HY	FAU	12MR	0.74
Phosphotungstic acid (HPW)	-	Non-porous	-

Table R1. The structural information of the samples used in adsorption-desorption experiments.

Figure R1. Confinement effect on the adsorption kinetics and thermodynamics of light olefins within materials with different pore sizes. *Top row:* Schematic representation of the structures of the investigated materials—SSZ-13 (8MR), ZSM-5 (10MR), HY (12MR), and HPW (non-porous solid acid)—arranged in order of increasing pore size. *Bottom row:* Adsorption results for C_2H_4 , C_3H_6 , and $n-C_4H_8$ on each material, showing the relationships between the adsorption rate constant (k_a) and equilibrium constant (K_A) with proton affinity. It is clear that as the pore size increases, the confinement-induced reversal between k_a and K_A progressively diminishes and ultimately disappears.

We first conducted experiments on SSZ-13, a representative small-pore CHA-type zeolite featuring a three-dimensional pore system with 8-membered-ring windows (8MR, 0.38 nm, Figure R2a), which are smaller than those of ZSM-5 (10 MR, 0.51×0.55 and 0.53×0.56 nm). Single-nanoparticle kinetic measurements revealed that k_a maintains a negative correlation

with the proton affinity of the three olefins, whereas K_A shows a positive correlation with proton affinity (Figure R2g). Notably, the adsorption kinetic of $n\text{-C}_4\text{H}_8$ was markedly suppressed, despite its highest K_A among the three adsorbates. These results further support our hypothesis that confinement within the zeolite framework is the dominant factor governing the adsorption kinetics of the olefins.

Figure R2. (a) CHA topology of SSZ-13 zeolite. (b-d) The normalized optical intensity curves of the three olefin molecules on single SSZ-13 nanoparticles during an adsorption-desorption cycle. The gray line represents the result obtained from each SSZ-13 nanoparticle. The color line represents the average signal of 37 SSZ-13 nanoparticles in the field of view. (e-f) Statistical histograms of k_a and K_A values for three olefins measured on each SSZ-13 particle. (g) The experimentally determined k_a (left axis) and K_A values (right axis) of three olefins from (e) as a function of their theoretically calculated protonation energies.

We then conducted experiments on HY nanoparticles, a representative FAU-type zeolite containing a three-dimensional pore structure of 12MR channels with a diameter of about 0.74nm (Figure R3a), which are larger than those in ZSM-5. As expected, the results showed that not only K_A but also k_a increases with the proton affinity of the studied olefins, suggesting that the previously observed reversal in adsorption kinetics is significantly weakened in this larger-pore environment (Figure R3g). This point is further confirmed by the result obtained on phosphotungstic acid (HPW), a classic non-porous solid acid, where both k_a and K_A also follow a positive correlation with proton affinity—consistent with fundamental acid-base theory that adsorption behavior is determined by intrinsic molecular interactions (Figure R4).

Figure R3. (a) FAU topology of HY zeolite. (b-d) The normalized optical intensity curves of the three olefin molecules on single HY nanoparticles during an adsorption-desorption cycle. The gray line represents the result from a HY nanoparticle. The color line represents the average signal of 125 HY nanoparticles in the field of view. (e-f) Statistical histograms of k_a and K_A values of three olefins measured on each HY nanoparticle. (g) The experimentally determined k_a (left axis) and K_A values (right axis) of three olefins from (e) as a function of their theoretically calculated protonation energies.

Figure R4. (a) Chemical structure of HPW. (b-d) The normalized optical intensity curves of the three olefin molecules on single HPW nanoparticles during an adsorption-desorption cycle. The gray line represents the result from a HPW nanoparticle. The color line represents the average signal of 42 HPW nanoparticles. (e-f) Statistical histograms of k_a and K_A values of three olefins measured on each HPW nanoparticle. (g) The experimentally determined k_a (left axis) and K_A values (right axis) of three olefins from (e) as a function of their theoretically calculated protonation energies.

Taken together, the adsorption results from SSZ-13, ZSM-5, HY, and HPW establish a clear structure–property relationship between pore confinement and adsorption kinetics. In the smallest-pore SSZ-13 (8MR), k_a shows a strong inverse correlation with proton affinity, whereas K_A increases with proton affinity, resulting in a pronounced reversal between adsorption

kinetics and thermodynamics driven by confinement. This reversal persists in ZSM-5 (10MR), consistent with its moderately larger pores. In contrast, when moving to HY (12MR) or HPW (a non-porous solid acid), both k_a and K_A revert to being governed by intrinsic molecular interaction strength (proton affinity), and the reversal disappears. The systematic comparison not only supports our hypothesis but also reveals the broad applicability of confinement-dependent adsorption behavior of olefins across diverse zeolite frameworks.

Relevant results and discussion have been included in the revised manuscript (Page 10, Figure 5) and supplementary information (Table S2).

Some aspects of the Arrhenius analysis are not entirely clear. The authors point out that with increasing affinity, both E_a and E_d increase. Under fixed partial pressure, exothermic adsorption should normally lead to lower equilibrium coverage at higher temperatures, so the plateau signal would be expected to vary with T . In Figures 4a, S15a, and S16a, the maximum responses (R_{max}) at different temperatures were pretty close. Is this because of the normalization or because the reaction equilibrium is not temperature-dependent (entropy-driven reaction)?

Response: We appreciate the reviewer's insightful comment that helped us clarify the temperature-dependent adsorption behavior of olefins on the same single zeolite nanoparticles and strengthen the rigor of our Arrhenius analysis. We agree that for an exothermic adsorption process, the equilibrium coverage generally decreases with increasing temperature. The nearly constant adsorption amounts observed at different temperatures in Figures 4a, S15a, and S16a, however, are primarily due to the relatively narrow experimental temperature range, in which the small changes in coverage are within the measurement error.

To verify this, we conducted additional C_2H_4 adsorption experiments on ZSM-5 over a broader temperature range. These results clearly show that the equilibrium adsorption amount decreased by approximately 25 % for a single nanoparticle when the temperature was increased from 25 °C to 200 °C (Figures R5a and R5b), consistent with the statistical results from 156 particles (Figure R5c). In contrast, within the 25-105 °C range used in the original experiments, the measured equilibrium adsorption amount remained nearly constant, indicating that the change in coverage is very small and below our measurement resolution.

Importantly, although the equilibrium adsorption amount among measurements could not be precisely resolved within this narrow temperature range, the measured kinetics remain robust, as k_a is determined from the temperature-dependent adsorption in each individual measurement. The temperatures selected (40 °C, 60 °C, 80 °C, 105 °C) were carefully chosen to optimize the accuracy of adsorption kinetic measurements. At high temperatures,

adsorption proceeds too rapidly and can exceed the temporal resolution of our method. Conversely, at low temperatures, adsorption becomes excessively slow, introducing measurement uncertainty. Moreover, as noted in the reviewer's following comment, elevated temperatures may also induce mild polymerization, coking, or gradual site deactivation, potentially affecting the reliability of the Arrhenius analysis. Therefore, the selected temperature range provides sufficient variation to reliably extract Arrhenius parameters while minimizing artifacts in the kinetic measurements.

Relevant results and discussion have been included in the revised manuscript (Page 9) and supplementary information (Page 19, Figure S18).

Figure R5. (a) The normalized optical intensity curves of C_2H_4 of single ZSM-5 nanoparticle from 25 $^{\circ}\text{C}$ to 200 $^{\circ}\text{C}$ during an adsorption-desorption cycle. (b) Plot of the equilibrium adsorption amount, extracted from panel (a), as a function of temperature. (c) The histograms of the equilibrium adsorption amount of 156 ZSM-5 particles under different temperatures.

Were the temperature series collected in a single continuous run or in multiple independent experiments? This information is important because, for C_2 olefins, high temperatures could lead to mild polymerization/coking or gradual site deactivation. Such effects may cause slow drifts in k_a and k_d , which would influence both the slopes (E_a , E_d) and intercepts (A) of the Arrhenius plots. A discussion of how such potential artifacts were controlled or excluded would strengthen the conclusions.

Response: We thank this reviewer for bringing this important point to our attention. In this study, to minimize the influence of particle-to-particle variability and ensure that changes

in k_a and k_d originate solely from temperature effects, the temperature-series measurements were carried out on the same ZSM-5 nanoparticles in a single experimental sequence. To further address potential concerns regarding mild polymerization, coking, or gradual site deactivation at elevated temperatures, new experiments and complementary analyses were carried out, as outlined below (Figure R6a). First, no appreciable changes were observed in the scattering intensity of the ZSM-5 nanoparticles before and after each measurement (Figure R6b), indicating that neither particle morphology nor optical properties were altered during the temperature-series experiments.

Second, the adsorption behaviors of the three olefins at the highest temperatures used for Arrhenius analysis were highly reproducible across repeated measurements. As shown in Figure R6f, both the extracted kinetic parameters and the equilibrium adsorption amounts were nearly identical when the adsorption–desorption sequence was repeated on the same single ZSM-5 nanoparticle, suggesting that no measurable site deactivation or polymerization occurred during the temperature-series experiments. This is consistent with the fact that the maximum temperature used for kinetic analysis (105 °C) is far below the temperatures reported in the literature (~250 °C), at which C₂ olefins begin to undergo significant coke formation on Brønsted acid sites.¹⁻³

Figure R6. (a) Experimental flowchart for adsorption measurements of three olefins. (b) The original optical

curves of a single ZSM-5 nanoparticle during the adsorption-desorption of three olefin molecules. (c-e) Representative single ZSM-5 particle adsorption-desorption kinetic curves for the cyclic adsorption process of three olefins. (f) The kinetic parameters extracted from (c-e). (g) The normalized optical intensity curves of the three olefin molecules on single ZSM-5 nanoparticles during an adsorption-desorption cycle. The gray lines represent the result obtained from a ZSM-5 nanoparticle, and the color lines represent the average response of all ZSM-5 nanoparticles.

Third, *in situ* IR spectroscopy on bulk ZSM-5 powder and confocal Raman measurements on single ZSM-5 nanoparticles directly confirm olefin adsorption without additional vibrational features associated with coking, or other side reactions. As shown in Figure R7, after 30 min of C_3H_6 adsorption at 100 °C, the differential IR spectrum of bulk ZSM-5—obtained by subtracting the pre-adsorption spectrum from the post-adsorption spectrum—exhibits a pronounced negative band at 3610 cm^{-1} , corresponding to the consumption of Brønsted acidic OH groups. Simultaneously, the occurrence of bands at 2955, 2930, and 2860 cm^{-1} ($\nu(CH_3)/\nu(CH_2)$) and 1470 cm^{-1} ($\delta(CH)$) indicates the formation of methyl/methylene species.^{4,5} These results demonstrate that C_3H_6 interacts primarily with Brønsted acid sites, without evidence of side reactions or contamination. Furthermore, we examined adsorption on a single ZSM-5 nanoparticle under identical conditions using confocal Raman microscopy. For the particle shown in Figure R8a, two peaks at 1340 and 1590 cm^{-1} emerge upon C_3H_6 adsorption, corresponding to C–O vibrational modes that confirm the formation of alkoxy-like species from chemisorption on Brønsted acid sites. This result further validates the adsorption process at the single-particle level. We note that the integrity and performance of the adsorption sites on ZSM-5 nanoparticles were also maintained through the experimental design. After each adsorption–desorption cycle, the sample was regenerated by heating to 200 °C under a nitrogen atmosphere for 10 minutes, ensuring the complete removal of residual adsorbates prior to the next measurement.

Figure R7. (a-b) The original *in-situ* IR adsorbance spectra before and after adsorption of C_3H_6 at 100 °C. (c-d) The differential IR spectrum of bulk ZSM-5—obtained by subtracting the pre-adsorption spectrum from the post-adsorption spectrum.

Figure R8. (a) Bright field image of single ZSM-5 nanoparticle on the gold film. (b) Raman spectra of the gold film (background, black line), and the label ZSM-5 particle in (a) before adsorption of C_3H_6 (blue line) as well as after adsorption of C_3H_6 (orange line).

Overall, these results confirm that the Arrhenius parameters obtained in this study are free from artifacts arising from thermal degradation or site deactivation, supporting the reliability of our kinetic analysis.

Relevant results and discussion have been included in the revised manuscript (Page 5 and Page 9) and supplementary information (Page 8 and Page 20, Figure S5 and S19) to strengthen the conclusion.

Reviewer #2 (Remarks to the Author):

Comments:

The manuscript entitled “Optical imaging of the intrinsic adsorption kinetics in single zeolite nanoparticles” reports a novel dark-field microscopy–based approach that enables quantitative measurement of intrinsic adsorption kinetics at the level of single ZSM-5 nanoparticles. The authors extracted adsorption/desorption rate constants and activation barriers, and further revealed a confinement-induced reversal of adsorption kinetics. This work presents a highly innovative methodology with important implications for adsorption studies in porous materials. However, the mechanistic interpretation and supporting evidence require significant strengthening. I therefore recommend major revision, with additional validation and a more rigorous discussion, before the manuscript can be considered for publication in Nature Communications.

Response: We appreciate this reviewer’s recognition of the novelty and significance of our methodology. To address this reviewer’s concern, we have conducted additional experiments and expanded our mechanistic discussion to provide stronger supporting evidence and clearer validation of our conclusions. Detailed point-by-point responses are provided below.

1. The change in scattering intensity is fully attributed to refractive index variation upon adsorption. Additional validation (e.g., *in situ* IR/Raman spectroscopy or mass spectrometry) would help exclude contributions from surface contamination, thermal effects, or other artifacts.

Response: We appreciate this reviewer for the insightful comment, which prompted us to systematically validate the origin of the scattering-intensity changes observed from ZSM-5 nanoparticles during the adsorption-desorption cycles. To ensure that the optical response is solely attributed to refractive-index variation induced by molecular adsorption, we have expanded the discussion and conducted additional complementary spectroscopic experiments. The supporting evidence is summarized below.

First, *in situ* IR spectroscopy on bulk ZSM-5 powder and confocal Raman measurements on single ZSM-5 nanoparticles both confirm olefin adsorption without additional vibrational features associated with coking, or other side reactions. As shown in Figure R1, after 30 min of C₃H₆ adsorption at 100 °C, the differential IR spectrum of bulk ZSM-5—obtained by subtracting the pre-adsorption spectrum from the post-adsorption spectrum—exhibits a pronounced negative band at 3610 cm⁻¹, corresponding to the consumption of Brønsted acidic OH groups. Simultaneously, the occurrence of bands at 2955, 2930, and 2860 cm⁻¹ ($\nu(\text{CH}_3)/\nu(\text{CH}_2)$) and 1470 cm⁻¹ ($\delta(\text{CH})$) indicates the formation of methyl/methylene species^{4,5}. These results demonstrate that C₃H₆ interacts primarily with Brønsted acid sites, without

evidence of side reactions or contamination. We further examined adsorption on a single ZSM-5 nanoparticle under identical conditions using confocal Raman microscopy. For the particle shown in Figure R10a, two peaks at 1340 and 1590 cm^{-1} emerge upon C_3H_6 adsorption, corresponding to C–O vibrational modes that confirm the formation of alkoxy-like species from chemisorption on Brønsted acid sites. This result further validates the adsorption process at the single-particle level.

Figure R9. (a-b) The original *in-situ* IR adsorbance spectra before and after adsorption of C_3H_6 at 100 °C. (c-d) The differential IR spectrum of bulk ZSM-5—obtained by subtracting the pre-adsorption spectrum from the post-adsorption spectrum.

Figure R10. (a) Bright field image of single ZSM-5 nanoparticle on the gold film. (b) Raman spectra of the gold film (background, black line), and the label ZSM-5 particle in (a) before adsorption of C_3H_6 (blue line) as well as after adsorption of C_3H_6 (orange line).

Second, the excellent recyclability of scattering intensity during repeated adsorption–desorption cycles provides additional mechanistic support. The scattering intensity of single

ZSM-5 nanoparticles could be reversibly switched between two distinct values across experiments (Figure R11). Alternative mechanisms such as thermal decomposition, carbon/pollutant deposition, or surface contamination would result in irreversible changes, contradicting the observed reversible behavior.

Figure R11. (a) Experimental flowchart for adsorption measurements of three olefins. (b) The original optical curves of a single ZSM-5 nanoparticle during the adsorption-desorption of three olefin molecules.

Third, although the adsorption/desorption of olefins is accompanied by a reversible thermal effect, the temperature change should follow the trend of the adsorption/desorption rate. Taking the adsorption process as an example, the surface temperature is expected to rise as adsorption proceeds and then gradually return to its initial baseline once equilibrium is reached due to heat dissipation (Fig. R12a). This characteristic transient heating profile is inconsistent with the steady equilibrium signal observed in the optical curves (Fig. R12b). Therefore, thermal effects associated with adsorption heat cannot account for the observed optical response.

Figure R12. (a) Schematic diagram of temperature changes during adsorption and desorption processes. (b) The Normalized optical intensity curve of a single ZSM-5 nanoparticle during an adsorption-desorption cycle of C_3H_6 .

Collectively, these complementary spectroscopic analyses, reversibility tests, and control experiments provide comprehensive and conclusive evidence that the observed scattering-

intensity change arises exclusively from refractive-index variation induced by molecular adsorption.

Relevant results and discussion have been included in the revised manuscript (Page 5 and Page 9) and supplementary information (Page 8 and Page 20, Figure S5 and S19) to validate the origin of the scattering-intensity changes observed from ZSM-5 nanoparticles during the adsorption–desorption cycles.

2. The pseudo-first-order reversible model may oversimplify the adsorption process. The possibility of multi-site adsorption, diffusion–reaction coupling, or site heterogeneity should be more thoroughly discussed, along with the limitations of the adopted model.

Response: We appreciate the reviewer’s concern regarding the simplicity of the pseudo-first-order reversible model in describing the real adsorption process. We agree that factors such as diffusion-reaction coupling, multi-site adsorption, and site heterogeneity may contribute to the overall adsorption kinetics and therefore warrant careful consideration. In this manuscript, we adopted the pseudo-first-order reversible model because it provides a physically meaningful and mathematically tractable framework for interpreting the transient adsorption profiles under our experimental conditions.

First, the interaction between the olefin molecules and the Brønsted acid sites in ZSM-5 constitutes the rate-determining step, while contributions from intracrystalline diffusion are negligible. This conclusion is supported by the particle size-independent adsorption rate constants (Figure S11), which rule out significant diffusion limitations.

Figure R13. (a) Pseudo-first-order kinetic fitting for NH₃ adsorption–desorption process. (b) Bi-exponential kinetic fitting for NH₃ adsorption–desorption process.

Second, the weakly basic olefins studied here adsorb predominantly on the strongest Brønsted acid sites of the zeolite; as a result, a single dominant class of adsorption sites governs the observed kinetics. Under such conditions, a pseudo-first-order reversible model is

both appropriate and sufficient. We note that this situation differs for stronger bases such as NH_3 , which can interact with both strong and weak acid sites. In those cases, a single first-order kinetic function would not adequately capture the system behavior, and a multi-site kinetic model—typically formulated as the sum of two first-order processes—would be required to describe the distinct adsorption pathways (Figure R13b).

We acknowledge that even within the category of “strong Brønsted acid sites”, subtle variations in acid strength may exist, which could introduce minor broadening in the kinetic profiles. However, such fine-scale heterogeneity cannot be resolved with the current temporal and spatial resolution of our single-particle measurement technique. Despite this limitation, the pseudo-first-order reversible model successfully captures the dominant kinetic behavior and allows for reliable comparison of adsorption kinetics across different adsorbates and confinement environments.

In the revised supplementary information, we have expanded the discussion to clarify both the rationale for using this model and its inherent limitations (Page 12).

3. The claim that confinement effects dominate adsorption kinetics is mainly supported by activation energy statistics and qualitative reasoning. The authors should consider providing additional support, such as molecular simulations (MD/DFT) or complementary spectroscopic evidence. Alternative explanations, such as acid site distribution or channel heterogeneity, should also be discussed.

Response: We appreciate the reviewer’s valuable comments. We agree that molecular simulations (MD/DFT) would provide complementary support for the mechanistic interpretation; however, we do not have the capability to perform such simulations at this stage. Instead, we have accordingly carried out systematic adsorption experiments on zeolites with different pore sizes and different acid site densities, and the result further supports our conclusion that confinement within the zeolite framework is the dominant factor governing the reversal trend between adsorption kinetics and proton affinity of the olefins.

On the one hand, the observed reversal of adsorption kinetics is strongly dependent on the pore size of zeolites, supporting the dominant role of confinement effects. We have conducted new experiments to systematically examine the adsorption behaviors of ethene (C_2H_4), propene (C_3H_6), and n-butene ($n\text{-C}_4\text{H}_8$) on zeolites with a range of pore sizes, including those both smaller and larger than the zeolite investigated in the original manuscript (Table R2). An overview of the results is presented in Figure R14, where the reversal between adsorption kinetics (k_a) and thermodynamics (K_A) becomes progressively weaker as the pore

size increases. A detailed discussion is provided below.

Sample	Framework Type	Featured number of rings (MR)	Pore size (nm)
SSZ-13	CHA	8 MR	0.38
ZSM-5	MFI	10 MR	0.51×0.55 0.53×0.56
HY	FAU	12MR	0.74
Phosphotungstic acid (HPW)	-	Non-porous	-

Table R2. The structural information of samples used in adsorption-desorption experiments.

Figure R14. Confinement effect on the adsorption kinetics and thermodynamics of light olefins within materials with different pore sizes. *Top row:* Schematic representations of the structures of the investigated materials—SSZ-13 (8MR), ZSM-5 (10MR), HY (12MR), and HPW (non-porous solid acid)—arranged in order of increasing pore size. *Bottom row:* Adsorption results for C₂H₄, C₃H₆, and n-C₄H₈ on each material, showing the relationships between the adsorption rate constant (k_a) and equilibrium constant (K_A) with proton affinity. It is clear that as the pore size increases, the confinement-induced reversal between k_a and K_A progressively diminishes and ultimately disappears.

We first conducted experiments on SSZ-13, a representative small-pore CHA-type zeolite featuring a three-dimensional pore system with 8-membered-ring windows (8MR, 0.38 nm, Figure R15a), which are smaller than those of ZSM-5 (10 MR, 0.51×0.55 and 0.53×0.56 nm). Single-nanoparticle kinetic measurements revealed that k_a maintains a negative correlation with the proton affinity of the three olefins, whereas K_A shows a positive correlation with proton affinity (Figure R15g). Notably, the adsorption kinetic of n-C₄H₈ was markedly suppressed, despite its highest K_A among the three adsorbates. These results support our hypothesis that confinement within the zeolite framework is the dominant factor governing the adsorption kinetics of the olefins.

Figure R15. (a) CHA topology of SSZ-13 zeolite. (b-d) The normalized optical intensity curves of the three olefin molecules on single SSZ-13 nanoparticles during an adsorption-desorption cycle. The gray line represents the result on each SSZ-13 nanoparticle. The color line represents the average signal of 37 SSZ-13 nanoparticles in the field of view. (e-f) Statistical histograms of k_a and K_A values of three olefins measured on each SSZ-13 particle. (g) The experimentally determined k_a (left axis) and K_A values (right axis) of three olefins from (e) as a function of their theoretically calculated protonation energies.

We then conducted experiments on HY nanoparticles, a representative FAU-type zeolite containing a three-dimensional pore structure of 12MR channels with a diameter of about 0.74nm (Figure R16a), which are larger than those in ZSM-5. As expected, the results showed that not only K_A but also k_a increases with the proton affinity of the studied olefins, suggesting that the previously observed reversal in adsorption kinetics is significantly weakened in this larger-pore environment (Figure R16g). This point is further confirmed by the result obtained on phosphotungstic acid (HPW), a classic non-porous solid acid, where both k_a and K_A also follow a positive correlation with proton affinity—consistent with fundamental acid-base theory that adsorption behavior is governed by intrinsic molecular interactions (Figure R17).

Figure R16. (a) FAU topology of HY zeolite. (b-d) The normalized optical intensity curves of the three olefin

molecules on single HY nanoparticles during an adsorption-desorption cycle. The gray line represents the result from a HY nanoparticle. The color line represents the average signal of 125 HY nanoparticles in the field of view. (e-f) Statistical histograms of k_a and K_A values of three olefins measured on each HY nanoparticle. (g) The experimentally determined k_a (left axis) and K_A values (right axis) of three olefins from (e) as a function of their theoretically calculated protonation energies.

Figure R17. (a) Chemical structure of HPW. (b-d) The normalized optical intensity curves of the three olefin molecules on single HPW nanoparticles during an adsorption-desorption cycle. The gray line represents the result from a HPW nanoparticle. The color line represents the average signal of 42 HPW nanoparticles. (e-f) Statistical histograms of k_a and K_A values of three olefins measured on each HPW nanoparticle. (g) The experimentally determined k_a (left axis) and K_A values (right axis) of three olefins from (e) as a function of their theoretically calculated protonation energies.

On the other hand, a consistent reversal dependence of adsorption kinetics on proton affinity was observed in zeolites with the same topological framework but different acid site densities. To investigate the effect of acid site distribution on adsorption kinetics, we conducted experiments on ZSM-5 with varying Si/Al ratios (15, 100, and 400) to decouple the effects of pore confinement and acid site density. As shown in Figure R18, a larger K_A was observed for ZSM-5 with a lower Si/Al ratio, as a result of stronger interactions between C_3H_6 and the more abundant acid sites. Meanwhile, the negative correlation between k_a and proton affinity persisted across all Si/Al ratios, demonstrating that acid site density has a negligible influence on the observed reversal in adsorption kinetics.

Figure R18. (a-c) The relationships between k_a and K_A with proton affinity for C_2H_4 , C_3H_6 and $n-C_4H_8$ on ZSM-5 with Si/Al ratios of 15, 100, and 400. (d-e) The summarized trends of k_a and K_A as a function of the proton affinity across varying Si/Al ratios.

In summary, our systematic experiments demonstrate that the observed reversal between adsorption kinetics and thermodynamics in zeolites is primarily governed by pore confinement effects rather than variations in acid site density. This conclusion is supported by measurements across zeolites with different pore sizes (SSZ-13, ZSM-5, HY) and varying Si/Al ratios. These findings confirm the robustness of our single-particle optical approach for probing intrinsic adsorption kinetics and provide new mechanistic insights into the role of confinement in zeolite catalysis.

Relevant results and discussion have been included in the revised manuscript (Page 10, Figure 5) and supplementary information (Page 21 and Page 23, Figure S20 and Table S2) to validate the origin of the observed reversal between adsorption kinetics and thermodynamics in zeolites.

4. About the generality of the conclusions: The study is limited to ZSM-5 and three light olefins. The applicability of the findings to other adsorbates or zeolite frameworks remains unclear. The authors should either provide broader data or explicitly delimit the scope of their conclusions.

Response: We appreciate this reviewer's constructive suggestion to further assess the generality of our conclusions. To address this point, we have expanded our study to additional zeolite frameworks and adsorbates.

First, we examined the adsorption kinetics of the three olefins in additional zeolite

framework, SSZ-13 with a CHA topology (Figure R19). The confinement-induced reversal between adsorption kinetics and thermodynamics persists across these different pore topologies, demonstrating that the phenomenon is not limited to ZSM-5. Moreover, based on the discussion related to Question #3, we believe that pore size, rather than the specific zeolite framework, plays a direct and decisive role in governing the observed kinetic reversal.

Figure R19. (a) CHA topology of SSZ-13 zeolite. (b) The normalized optical intensity curves of the three olefin molecules on single SSZ-13 nanoparticle during an adsorption-desorption cycle at 130 °C. (c) The experimentally determined k_a (left axis) and K_A values (right axis) of three olefin molecules as a function of their theoretically calculated protonation energies.

Second, we investigated another important class of adsorbates—light alcohols (methanol, ethanol, and n-propanol)—whose molecular sizes are comparable to those of the light olefins but whose adsorption involves distinct hydrogen-bonding and acid–base interactions. Because these alcohols are liquid at room temperature, their vapor-phase adsorption on ZSM-5 was measured via nitrogen bubbling. As shown in Figure R20, these alcohols also exhibit the same confinement-dependent kinetic reversal, confirming that the observed behavior is not specific to olefins.

Figure R20. (a) MFI topology of ZSM-5 zeolite. (b) The normalized optical intensity curves of the three alcohol molecules on single ZSM-5 nanoparticle during an adsorption-desorption cycle at 100 °C. (c) The experimentally determined k_a (left axis) and K_A values (right axis) of three alcohol molecules as a function of their theoretically calculated protonation energies.

Collectively, the additional experiments across different adsorbates and zeolite frameworks substantiate the broader applicability of our conclusions determined from the

initial study focused on the adsorption of light olefins on ZSM-5 nanoparticles.

Relevant results and discussion have been included in the revised manuscript (Page 10, Figure 5) and supplementary information (Page 22, Figure S21) to demonstrate the broad applicability of the observed reversal between adsorption kinetics and thermodynamics in zeolites.

Reviewer #3 (Remarks to the Author):

Comments:

Response: We appreciate the opportunity to have our manuscript reviewed under the *Nature Communications* co-review initiative and thank the reviewer for their careful evaluation and constructive feedback.

Reviewer #4 (Remarks to the Author):

Comments:

The confinement effect has been well accepted in zeolite chemistry, as it can significantly influence the adsorption of adsorbates and overall catalytic performance in catalysis. In this work, Yi et al. correlate adsorption kinetics with the confinement effect in zeolites by employing optical imaging techniques to observe single ZSM-5 nanoparticles. By monitoring optical intensity changes, they assess the intrinsic adsorption and desorption kinetics. Overall, this is an interesting study; however, several major concerns should be addressed as follows:

Response: We appreciate this reviewer's valuable comments that help us to improve the quality of our manuscript. Detailed point-by-point responses are as follows.

1. There is a critical lack of clarity and completeness regarding the characterization of the zeolite used in this work. The author states that three distinct materials were employed: ZSM-5 (with Si/Al ratios of 50 and 400) and silicalite-1. However, the authors state that only "the ZSM-5 nanoparticles" were characterized (Fig. S1), without specifying which Si/Al ratio this refers to. Besides, the acidity of those three samples should be evaluated, for example, the Bronsted acid site and Lewis acid site. Probably, the solid-state NMR measurements can be conducted to provide

more detailed information.

Response: We sincerely apologize for the insufficient clarity and completeness in the description of the zeolite materials used in this work. We thank the reviewer for pointing out this issue that helps us to improve the rigor of the manuscript. Accordingly, we have comprehensively characterized all materials used, including their Si/Al ratios, crystal structures, and Brønsted acid site densities. Detailed results are discussed as follows.

First, the chemical composition of the ZSM-5 with different Si/Al ratios was determined by an inductively coupled plasma optical emission spectrum (ICP-OES). To avoid ambiguity, the ZSM-5 samples with different Si/Al ratios were named accordingly (e.g., ZSM-5-15, ZSM-5-50, ZSM-5-100, ZSM-5-400). As listed in Table R3, the ICP-OES results confirm that the measured Si/Al ratios are in good agreement with the values provided by the supplier.

	W(Si)%	W(Al)%	(Si/Al)_{ICP}
ZSM-5-15	40.4615	2.4604	15.8
ZSM-5-50	41.5125	0.7649	52.1
ZSM-5-100	41.9893	0.3753	107.5
ZSM-5-400	43.2910	0.1008	412.6
Silicalite-1	44.6671	0.0167	2569.5

Table R3. Si/Al ratios of all materials used in this work, determined by ICP-OES.

Second, the crystal structures of all materials used in this work were characterized by X-ray diffraction (XRD). The diffraction patterns of each material match well with the characteristic peaks reported in the literature for the corresponding MFI frameworks (Figure R21a),⁶ confirming that all samples possess the expected crystalline structure and phase purity.

Third, the acidity of all materials used in this work was evaluated by NH₃-TPD. As shown in Figure R21b, both the desorption temperature and the total amount of acid sites increase as the Si/Al ratio of ZSM-5 decreases. This trend is consistent with the expected increase in Brønsted acid site density and acid strength at lower Si/Al ratios, further confirming the reliability of the used materials and thus the corresponding conclusion.

Figure R21. Characterization results of the ZSM-5 particles used in this work. (a, c) XRD patterns. (b, d) NH₃-TPD spectra.

Relevant results and discussion have been included in the revised supplementary information (Figure S1, Figure S12, Figure S20 and Table S1). The Si/Al ratio of all ZSM-5 have been clarified in the revised manuscript and the supplementary information.

2. The authors state that the NH₃-TPD spectrum shows peaks "at around 200°C and 400°C". However, upon inspection of Fig. S1c, the second desorption peak appears to be located at a significantly different temperature (approximately 430°C). The text should be revised to accurately reflect the data presented in the figure. Besides, how much acid site in this zeolite? Is there any quantitative relationship between the acid site and the adsorbates adsorption?

Response: We appreciate this reviewer for pointing out this mistake. The temperatures of desorption peaks have been re-examined, and the maxima are located at 199 °C and 433 °C, respectively (Figure R22). The corresponding values have been corrected in the revised manuscript (Page 4) and supplementary information (Page 4, Figure S1).

Additionally, the acid site amounts were quantified from the peak areas of the NH₃-TPD spectrum, where the amount of weak and strong acid sites in this ZSM-5 sample with Si/Al ratio of 50 is determined to be 0.139 mmol/g and 0.093 mmol/g, respectively.

Figure R22. NH₃-TPD of ZSM-5 with Si/Al of 50.

Furthermore, it is evident that adsorbate adsorption varies significantly among different ZSM-5 nanoparticles, as reflected in the $\Delta I/I_0$ values during C₃H₆ adsorption (Figure R23). To further explore the relationship between acid site density and adsorbate adsorption, it would be ideal to measure acid site amounts at the single nanoparticle level. While NH₃-TPD provides reliable bulk measurements, it lacks the sensitivity and spatial resolution to resolve individual nanoparticles. Inspired by this technique, we employed the equilibrium adsorption amount of NH₃ ($\Delta I/I_{0, \text{NH}_3}$) as a proxy for the number of acid sites on single ZSM-5 nanoparticles. A positive correlation between the $\Delta I/I_{0, \text{NH}_3}$ (acid site amounts) and $\Delta I/I_{0, \text{C}_3\text{H}_6}$ (adsorbates adsorption amount) was found (Figure R23c), indicating that the adsorption capacity is strongly influenced by the local acid site density. By contrast, no dependence was found between C₃H₆ adsorption kinetics (k_a) and acid site amounts (Figure R23d), consistent with our conclusion that the observed confinement-induced kinetic reversal is governed primarily by pore structure rather than the absolute number of acid sites.

Figure R23. (a) The normalized optical intensity curves of 74 single ZSM-5 nanoparticles during an NH_3 adsorption-desorption cycle at 100 °C. (b) The normalized optical intensity curves of the same 74 single ZSM-5 nanoparticles during C_3H_6 adsorption-desorption cycle at 130 °C. (c) The statistical dependence of $\Delta I/I_0$ between NH_3 and C_3H_6 for the adsorption steps. (d) No significant correlation was observed between adsorption kinetics and adsorption capacity.

3. A significant discrepancy exists between Figure 1b and Supplementary Figure S5, which purportedly show the same phenomenon of the adsorption and desorption of C_3H_6 on ZSM-5. The magnitude of the optical intensity change (the y-axis value) is markedly different between the two figures. In Fig. 1b, the signal change appears to be approximately 0.05, while in Fig. S5 (yellow curve), it appears to be around 0.035.

Response: We thank the reviewer for bringing this inconsistency to our attention and apologize for the confusion caused by the discrepancy between Figure 1b and Figure S5. The difference in the magnitude of the optical intensity change arose because Figure 1b originally displayed representative data from a different ZSM-5 nanoparticle than the one shown in Figure S5. The optical response naturally varies from particle to particle due to structural heterogeneity, including differences in acid site amounts, as discussed in Q#2.

To ensure clarity and consistency throughout the manuscript, we have revised Figure 1b to use the same nanoparticle as in Figures S5 and S7. This correction ensures that all figures consistently represent the same adsorption-desorption behavior.

4. A critical clarification is required regarding the fundamental process under investigation. The manuscript consistently describes the interaction of C_3H_6 with Brønsted acid sites as a

"reaction" (e.g., in the context of Fig. 1b). However, the experimental data presented (the reversible optical intensity change in Fig. 1b and the subsequent desorption in TPD profiles) unequivocally characterize a physical/chemical adsorption event, not a chemical reaction. Besides, what's the effect of the Lewis acid site and the defect sites (in both the outside and inner surface) effect on the adsorption effect?

Response: We appreciate this reviewer's valuable comment, which highlights the need for precision in describing the fundamental process under investigation. We acknowledge that referring to the interaction of C₃H₆ with Brønsted acid sites as a "reaction" is misleading. The experimental data, including the reversible optical intensity changes in Fig. 1b and the desorption behavior in TPD measurement, clearly indicate that the process is a reversible adsorption event rather than a chemical reaction. Accordingly, we have revised the manuscript to consistently describe these interactions as "chemical adsorption".

Moreover, to identify the chemical adsorption process, *in situ* IR spectroscopy and confocal Raman microscopy were employed to characterize the bulk ZSM-5 powder and single ZSM-5 nanoparticle before and after C₃H₆ adsorption. As shown in Figure R24, after 30 min of C₃H₆ adsorption at 100 °C, the differential IR spectrum of bulk ZSM-5—obtained by subtracting the pre-adsorption spectrum from the post-adsorption spectrum—exhibits a pronounced negative band at 3610 cm⁻¹, corresponding to the consumption of Brønsted acidic OH groups. Simultaneously, the occurrence of bands at 2955, 2930, and 2860 cm⁻¹ ($\nu(\text{CH}_3)/\nu(\text{CH}_2)$) and 1470 cm⁻¹ ($\delta(\text{CH})$) indicates the formation of methyl/methylene species^{4,5}. In addition, for the particle shown in Figure R24, two Raman peaks at 1340 and 1590 cm⁻¹ emerge upon C₃H₆ adsorption, corresponding to C–O vibrational modes. These results provide direct evidence that C₃H₆ adsorption occurs mainly on Brønsted acid sites via chemical interaction.

Figure R24. (a-b) The original *in-situ* IR adsorbance spectra before and after adsorption of C₃H₆ at 100 °C. (c-d) The corresponding differential IR spectrum of bulk ZSM-5—obtained by subtracting the pre-adsorption spectrum from the post-adsorption spectrum.

Furthermore, to investigate the effects of Lewis acid sites and defect sites on adsorption, we employed an *in situ* NaOH etching experiment at the single-nanoparticle level. This treatment has two main effects: (1) partial extraction of framework silicon, generating defect sites, and (2) selective removal of extra-framework Al (EFAL) species, a major source of Lewis acid sites. The experimental protocol is illustrated in Figure R25a. The results before and after NaOH etching reveal heterogeneous responses among different nanoparticles, and representative adsorption–desorption kinetics for three ZSM-5 nanoparticles are shown in Figures R25b-d. Some particles showed increased uptake and faster kinetics, others exhibited increased uptake but slower kinetics, and a minority displayed decreased adsorption, which likely originated from severe silicon removal. The complexity of these results hinders a clear assessment of the individual contributions of Lewis acid sites and defect sites, since their effects are entangled in the observed adsorption behaviors. As we were unable to selectively and independently modulate these structural features, a definitive conclusion regarding their separate impacts on adsorption kinetics cannot be drawn at this stage.

Figure R25. (a) The experimental workflow of *in situ* NaOH etching experiment. (b-c) Representative adsorption-desorption kinetic curves of single ZSM-5 nanoparticles before (blue) and after (orange) NaOH etching.

Relevant results have been included in the revised manuscript (Page 5) and supplementary information (Page 8, Figure S5).

5. The data in Supplementary Figure S11d appears to directly contradict the interpretation provided for the ZSM-5 system and instead points towards a diffusion-controlled process for silicalite-1. The authors show a clear negative correlation between the adsorption rate constant of C_3H_6 and the particle size in silicalite-1. This is a classic signature of a diffusion-limited process: in larger particles, molecules have to travel a longer average path to access the internal pore network, leading to a slower observed adsorption rate.

Response: We apologize for the unclear description that may have caused confusion. We agree that the adsorption of C_3H_6 on Silicalite-1 is a diffusion-limited process, as evidenced by the clear particle-size dependence of the adsorption rates—larger Silicalite-1 particles require longer times to reach equilibrium. In contrast, adsorption on ZSM-5 does not exhibit noticeable particle-size dependence, and both adsorption and desorption rates are substantially slower than those observed for Silicalite-1 of comparable size. This comparison suggests that, in ZSM-5, the adsorption kinetics are dominated by the interactions between olefin molecules and the strong Brønsted acid sites rather than by intracrystalline diffusion, thereby providing a system in which the intrinsic interaction kinetics within zeolites can be directly studied—the focus of our manuscript.

Moreover, the key conclusion we aim to convey is that the interaction kinetics between olefin molecules and Brønsted acid sites are strongly influenced by the confinement effect

imposed by the microporous environment of ZSM-5. For instance, $n\text{-C}_4\text{H}_8$, being a stronger base than C_2H_4 , should exhibit stronger and faster interactions with the same acid sites, as confirmed by control experiments on phosphotungstic acid (HPW, with further details provided in the response to Q#8 below), a classic non-porous solid acid. However, this trend reverses in ZSM-5: the adsorption rate of C_2H_4 becomes even faster than that of $n\text{-C}_4\text{H}_8$. This reversal highlights the decisive role of pore confinement in governing intrinsic interaction kinetics within zeolites.

The relevant discussion has been reorganized and clarified in the revised supplementary information to avoid potential confusion (Page 15, Figure S12).

6. L242-245: The claim that the increase in optical intensity with molecular weight suggests "effective access" to active sites is not logically sound. The optical signal is primarily a function of the adsorbed mass. A heavier molecule will naturally produce a larger signal per molecule, regardless of diffusion or accessibility limitations. The authors should either provide additional data to support the accessibility claim (e.g., adsorption capacities) or rephrase the conclusion to state that the data confirms adsorption occurs, with the signal intensity being mass-dependent.

Response: We thank the reviewer for this comment. We agree that heavier molecules adsorbed on the acid sites of ZSM-5 inherently produce larger optical signals due to the mass dependence of the scattering response. In our measurements, the $\Delta I/I_0$ values of the three olefins on the same ZSM-5 nanoparticle not only increase with molecular weight but also exhibit a clear linear dependence (Figure R26). This linearity indicates that the number of adsorbed molecules (or equivalently, the number of accessed acid sites) is comparable for the three olefins, and that the differences in signal magnitude arise primarily from their molar masses rather than from variations in accessibility. Accordingly, we conclude that all three studied olefins are able to access and interact with the Brønsted acid sites under the experimental conditions.

The relevant description has been reorganized in the revised manuscript to avoid potential confusion (Page 7).

Figure R26. (a) The optical adsorption-desorption curves of the three olefin molecules on a ZSM-5 nanoparticle. (b) The linear dependence between the observed $\Delta I/I_0$ and the corresponding molar weight of the three olefins.

7. Regarding the analysis in Fig. 3c, the manuscript states that the equilibrium constants (K_A) were determined from the "statistical linear dependence" between elementary adsorption and desorption rate constants. However, the actual K_A values and the crucial statistical parameter (e.g., the coefficient of determination, R^2) for this linear fit are not provided either in the figure, the caption, or the main text.

Response: We apologize for the omission of crucial statistical parameters in the original manuscript. The supplementary parameters, including the equilibrium constants (K_A) and the coefficients of determination (R^2) for the linear fits, have been added to the figure caption in the revised manuscript (Figure 3).

Figure R27. The statistical linear dependence between the rate constants for the elementary adsorption and desorption steps of each olefin molecule, where yellow, orange, and purple dots represent the results of C_2H_4 , C_3H_6 and $n-C_4H_8$, respectively. $R^2_{C_2H_4} = 0.84$, $R^2_{C_3H_6} = 0.97$, $R^2_{n-C_4H_8} = 0.95$, $K_{A\ C_2H_4} = 1.21 \times 10^{-2} \text{ kPa}^{-1}$, $K_{A\ C_3H_6} = 2.38 \times 10^{-2} \text{ kPa}^{-1}$, $K_{A\ n-C_4H_8} = 1.61 \times 10^{-1} \text{ kPa}^{-1}$.

8. In this work, the authors claim that it is the confinement effect but the interaction strength works as the main factor to control the adsorption kinetics of adsorbates. Personally, I don't agree this statement, Confinement is not a standalone steric effect but a holistic one that includes how acid strength manifests in tight spaces. In other words, confinement effect is an overall effect, which contains the contribution of acid strength for adsorbates adsorption. Besides, only one type zeolite was adopted in this work, the different topology zeolite with different channel size should be involved for a systemic comparison.

Response: We thank the reviewer for this insightful comment. We fully agree that confinement is not a standalone steric effect but a holistic phenomenon, where the spatial constraints of the pore influence how the acid strength of Brønsted sites is expressed and

experienced by adsorbates. Under tight confinement, such as in the 10-membered ring channels of ZSM-5, the pore geometry becomes the dominant factor controlling adsorption kinetics, overriding the trends predicted solely from intrinsic gas-phase proton affinity. In this sense, the confinement effect inherently includes the influence of acid–adsorbate interactions.

To further validate this conclusion, we have conducted new experiments to systematically examine the adsorption behaviors of ethene (C_2H_4), propene (C_3H_6), and n-butene ($n-C_4H_8$) on zeolites with a range of pore sizes, including those both smaller and larger than the zeolite investigated in the original manuscript (Table R4). An overview of the results is presented in Figure R28, where the reversal between adsorption kinetics (k_a) and thermodynamics (K_A) becomes progressively weaker as the pore size increases. A detailed discussion is provided below.

Sample	Framework Type	Featured number of rings (MR)	Pore size (nm)
SSZ-13	CHA	8 MR	0.38
ZSM-5	MFI	10 MR	0.51×0.55 0.53×0.56
HY	FAU	12MR	0.74
Phosphotungstic acid (HPW)	-	Non-porous	-

Table R4. The structural information of the samples used in adsorption-desorption experiments.

Figure R28. Confinement effect on the adsorption kinetics and thermodynamics of light olefins within materials with different pore sizes. *Top row:* Schematic representations of the structures of the investigated materials—SSZ-13 (8MR), ZSM-5 (10MR), HY (12MR), and HPW (non-porous solid acid)—arranged in order of increasing pore size. *Bottom row:* Adsorption results for C_2H_4 , C_3H_6 , and $n-C_4H_8$ on each material, showing the relationships between the adsorption rate constant (k_a) and equilibrium constant (K_A) with proton affinity. It is clear that as the pore size increases, the confinement-induced reversal between k_a and K_A progressively diminishes and ultimately disappears.

We first conducted experiments on SSZ-13, a representative small-pore CHA-type zeolite featuring a three-dimensional pore system with 8-membered-ring windows (8MR, 0.38 nm, Figure R29a), which are smaller than those of ZSM-5 (10 MR, 0.51×0.55 and 0.53×0.56 nm). Single-nanoparticle kinetic measurements revealed that k_a maintains a negative correlation with the proton affinity of the three olefins, whereas K_A shows a positive correlation with proton affinity (Figure R29g). Notably, the adsorption kinetic of n -C₄H₈ was markedly suppressed, despite its highest K_A among the three adsorbates. These results support our hypothesis that confinement within the zeolite framework is the dominant factor governing the adsorption kinetics of the olefins.

Figure R29. (a) CHA topology of SSZ-13 zeolite. (b-d) The normalized optical intensity curves of the three olefin molecules on single SSZ-13 nanoparticles during an adsorption-desorption cycle. The gray line represents the result on each SSZ-13 nanoparticle. The color line represents the average signal of all SSZ-13 nanoparticles in the field of view. (e-f) Statistical histograms of k_a and K_A values of three olefins measured on each SSZ-13 particle. (g) The experimentally determined k_a (left axis) and K_A values (right axis) of three olefins from (e) as a function of their theoretically calculated protonation energies.

We then conducted experiments on HY nanoparticles, a representative FAU-type zeolite containing a three-dimensional pore structure of 12MR channels with a diameter of about 0.74nm (Figure R30a), which are larger than those in ZSM-5. As expected, the results showed that not only K_A but also k_a increases with the proton affinity of the studied olefins, suggesting that the previously observed reversal in adsorption kinetics is significantly weakened in this larger-pore environment (Figure R30g). This point is further confirmed by the result obtained on phosphotungstic acid (HPW), a classic non-porous solid acid, where both k_a and K_A also follow a positive correlation with proton affinity—consistent with fundamental acid-base theory that adsorption behavior is governed by intrinsic molecular interactions (Figure R31).

Figure R30. (a) FAU topology of HY zeolite. (b-d) The normalized optical intensity curves of the three olefin molecules on single HY nanoparticles during an adsorption-desorption cycle. The gray line represents the result from a HY nanoparticle. The color line represents the average signal of 125 HY nanoparticles in the field of view. (e-f) Statistical histograms of k_a and K_A values of three olefins measured on each HY nanoparticle. (g) The experimentally determined k_a (left axis) and K_A values (right axis) of three olefins from (e) as a function of their theoretically calculated protonation energies.

Figure R31. (a) Chemical structure of HPW. (b-d) The normalized optical intensity curves of the three olefin molecules on single HPW nanoparticles during an adsorption-desorption cycle. The gray line represents the result from a HPW nanoparticle. The color line represents the average signal of 42 HPW nanoparticles. (e-f) Statistical histograms of k_a and K_A values of three olefins measured on each HPW nanoparticle. (g) The experimentally determined k_a (left axis) and K_A values (right axis) of three olefins from (e) as a function of their theoretically calculated protonation energies.

Relevant results and discussion have been included in the revised manuscript (Page 10, Figure 5) and supplementary information (Figure S20 and Table S2).

Minor error:

1. In Supplementary Figure S4, the unit for milliliter is written as "ml". The correct SI symbol should be "mL".

Response: We thank the reviewer for pointing this error out. It has been corrected in the revised Supplementary materials.

References

1. Zabihpour, A., Ahmadpour, J., Yaripour, F. Strategies to control reversible and irreversible deactivation of ZSM-5 zeolite during the conversion of methanol to propylene (MTP): A review. *Chem. Eng. Sci.* **273**, 118639 (2023).
2. Wang, F., Luo, M., Xiao, W. D. Catalytic Performance and Coking Behavior of a Submicron HZSM-5 Zeolite in Ethanol Dehydration. *Chin. J. Chem.* **29**, 1326-1334 (2011).
3. Tabernilla, Z., Ateka, A., Aguayo, A. T., Epelde, E. Unraveling the Role of HZSM-5 Zeolite Catalyst Acid Properties in Fuel Production and Coke Deactivation in Low-Pressure Ethylene Oligomerization. *Energy Fuels.* **39**, 8639-8651 (2025).
4. Spoto, G., Bordiga, S., Ricchiardi, G., Scarano, D., Zecchina, A., Borello, E. IR STUDY OF ETHENE AND PROPENE OLIGOMERIZATION ON H-ZSM-5 - HYDROGEN-BONDED PRECURSOR FORMATION, INITIATION AND PROPAGATION MECHANISMS AND STRUCTURE OF THE ENTRAPPED OLIGOMERS. *J. Chem. Soc.-Faraday Trans.* **90**, 2827-2835 (1994).
5. Yamazaki, H., Yokoi, T., Tatsumi, T., Kondo, J. N. Ethene oligomerization on H-ZSM-5 in relation to ethoxy species. *Catalysis Science & Technology.* **4**, 4193-4195 (2014).
6. Zeng, B. J. et al. Directional Transport in Hierarchically Aligned ZSM-5 Zeolites with High Catalytic Activity. *J. Am. Chem. Soc.* **146**, 33423-33433 (2024).

Responses to Reviewers' Comments

Reviewer #1 (Remarks to the Author):

Comments: The revision has addressed all questions with new data and additional discussions. I think it is ready to be accepted.

Response: We sincerely thank the reviewer for their positive evaluation of our revisions and for recommending the manuscript for publication in *Nature Communications*. We greatly appreciate the reviewer's time, effort, and constructive feedback, which have helped improve the quality and clarity of our manuscript.

Reviewer #2 (Remarks to the Author):

Comments: I appreciate the authors' thorough and detailed responses to my original comments. The authors have made substantial improvements to the manuscript. The additional experiments, particularly the systematic comparison across zeolites with different pore sizes (SSZ-13, ZSM-5, HY) and phosphotungstic acid, provide compelling evidence and significantly strengthen the manuscript. However, before final acceptance, I would like the authors to address the following remaining concerns:

Response: We appreciate this reviewer's valuable comments, which have greatly helped us improve the manuscript. We are glad to hear that our previous revision addressed most of the reviewer's concerns. We will make every effort to clarify the remaining concerns in this revision.

1. NaOH etching experiment design. The authors performed NaOH etching experiments to investigate the effects of Lewis acid sites and defect sites. However, I noticed that after alkaline treatment and water washing, the samples were directly dried and measured without ion-exchange and calcination steps. This means the measured samples are Na-form ZSM-5 rather than H-form ZSM-5, as the Brønsted acid sites have been occupied by Na⁺ cations. The correct procedure should include: NaOH etching → water washing → NH₄⁺ ion exchange → calcination → H-ZSM-5. The current experimental design confounds the effects of acid site loss with defect introduction, which likely explains the complex and inconclusive results. I suggest the authors provide control experiments with proper ion-exchange and calcination steps.

Response: We thank the reviewer for the constructive comments and for pointing out the importance of ion-exchange and calcination in the NaOH etching experiments. Following the

reviewer's suggestion, we repeated the experiments using the corrected procedure, including NH_4^+ ion exchange and subsequent calcination (Fig. R1a), to restore the H-form ZSM-5 after alkaline treatment.

The revised experiments show that, after NaOH etching followed by ion exchange and calcination, the change in adsorption capacity decreases systematically for all investigated nanoparticles (Fig. R1b), which is consistent with partial framework silicon extraction and the associated loss or restructuring of adsorption-active sites induced by alkaline treatment. Despite this uniform decrease in adsorption capacity, the adsorption-desorption kinetics remain highly heterogeneous across individual nanoparticles, with no clear or systematic trend emerging (Fig. R1c). These results indicate that, although the additional control experiments improve experimental rigor, the effects of NaOH etching on adsorption behavior still remain complex, particularly with respect to adsorption-desorption kinetics, which is the key focus of this work. This complexity is expected, as NaOH treatment simultaneously induces framework desilication and acid-site modification, making it challenging to decouple their respective contributions using this approach alone. Importantly, this coupling does not affect the main conclusions of the manuscript, which are based on consistently observed adsorption kinetic trend across different olefin species at the single nanoparticle level and are independent of the NaOH etching experiments.

Figure R1. (a) The experimental workflow of *in-situ* NaOH etching experiment. (b) The histograms of $\Delta I/I_0$ of

74 ZSM-5 particles before (blue) and after (orange) NaOH etching. (c) Representative adsorption-desorption kinetic curves of three representative ZSM-5 nanoparticles before (blue) and after (orange) NaOH etching.

2. Source of protonation energy data. The authors use "theoretically calculated protonation energies" as a key parameter to correlate adsorption kinetics and thermodynamics (Figure 3d, Figure 5c-d, etc.). However, the source and calculation method of these values are not specified. The authors should note that in confined zeolite environments, actual protonation/adsorption energies may significantly deviate from gas-phase values due to geometric constraints, van der Waals interactions, and local acid site environments. I recommend the authors clearly state the source and calculation method of the protonation energy data. If possible, provide DFT calculations of adsorption energies and transition state barriers within the confined zeolite channels, which would greatly strengthen the mechanistic interpretation.

Response: We appreciate the reviewer for raising this important point regarding the source and applicability of the protonation energy data. We agree that molecular simulations (MD/DFT) would provide more accurate protonation and adsorption energies for mechanistic interpretation; however, we do not have the capability to perform such simulation at this stage. In the revised manuscript, we have clearly stated that the protonation energies used in Figures 3d and 5c-d are theoretically calculated gas-phase values obtained from the National Institute of Standards and Technology (NIST, <https://webbook.nist.gov/chemistry/>), with the corresponding references explicitly added.

Although absolute protonation energies depend to some extent on the computational method and geometric constraints, the relative protonation strength of the three olefins is invariant and well established (J. Chem. Theory Comput. 2014, 10, 3123-3128). Importantly, this ordering is independently validated by experimental equilibrium association constants (K_A), which scale with adsorption affinity and exhibit the same trend across zeolites with distinct pore sizes (SSZ-13, ZSM-5, and HY) and phosphotungstic acid, as demonstrated in our previous revision and summarized in Fig. R2.

Figure R2. Confinement effect on the adsorption kinetics and thermodynamics of light olefins within materials with different pore sizes. *Top row:* Schematic representation of the structures of the investigated materials—SSZ-13 (8MR), ZSM-5 (10MR), HY (12MR), and HPW (non-porous solid acid)—arranged in order of increasing pore size. *Bottom row:* Adsorption results for C₂H₄, C₃H₆, and n-C₄H₈ on each material, showing the relationships between the adsorption rate constant (k_a) and equilibrium constant (K_A) with proton affinity. It is clear that as the pore size increases, the confinement-induced reversal between k_a and K_A progressively diminishes and ultimately disappears.

Moreover, the reverse trend observed between the equilibrium association constants (K_A) and adsorption rate constants (k_a) of the three olefins on the very same nanoparticle provides direct evidence supporting our conclusion. This further confirms that gas-phase protonation energies serve as a reliable comparative descriptor of the *intrinsic* interaction strength between olefins and acid sites in these systems.

Relevant references and discussion have been included in the revised manuscript (Page 8, 12 and 13).

3. Discussion of crystal diversity and activation energy distribution. The histograms in Figure 4d-e show substantial broadening in the activation energy distributions across 50 ZSM-5 particles (e.g., E_a for C₃H₆ ranges from ~25 to ~70 kJ/mol). This particle-to-particle heterogeneity is consistent with the "crystal diversity" phenomenon reported by Saint Remi et al. (Nat. Mater. 15, 401-406, Ref. 22). I suggest the authors discuss the possible origins of this activation energy broadening and analyze how this inter-particle heterogeneity affects the reliability of the conclusions. Once the above concerns are adequately addressed, I recommend acceptance of this manuscript for publication in Nature Communications.

Response: We thank the reviewer for highlighting this point. The significant broadening of the activation energy distributions reflects inter-particle heterogeneity within the same batch of

ZSM-5 particles, consistent with the "crystal diversity" phenomenon reported by Saint Remi et al. We would like to clarify that this heterogeneity does not compromise the reliability of our main conclusion—that the adsorption rates of light olefins exhibit a reversed trend relative to their gas-phase proton affinities. This is because our single-particle measurements compare the adsorption behaviors of the three light olefins within the same particle, and the reversed trend is consistently observed across all individual particles, with a representative particle shown in Fig. 3b.

Importantly, this heterogeneity also underscores the significance of our single-particle measurement strategy, as such trend may be masked in bulk measurements due to averaging effects. Regarding the possible origins of the observed heterogeneity, our preliminary analysis suggests it may arise from differences in defect densities and subtle variations in pore structure orientation among individual ZSM-5 crystals. While a comprehensive investigation of these factors is beyond the scope of the present work, we acknowledge the fundamental importance of this phenomenon and note that a detailed study is currently underway. We look forward to the reviewer's valuable comments on this topic in our future work.

Reviewer #3 (Remarks to the Author):

Comments:

Response: We appreciate the opportunity to have our manuscript reviewed under the *Nature Communications* co-review initiative and thank the reviewer for their careful evaluation and constructive feedback.

Reviewer #4 (Remarks to the Author):

Comments:

The authors have addressed my concerns, I am satisfied with the revisions and recommended its publication in Nature Communications.

Response: We sincerely thank the reviewer for their positive evaluation of our revisions and for recommending the manuscript for publication in *Nature Communications*. We greatly

appreciate the reviewer's time, effort, and constructive feedback, which have helped improve the quality and clarity of our manuscript.